**Registered report**

# Antibiotic use and survival from breast cancer: A population-based cohort study in England and Wales

Chris R. Cardwell [1] ✉, Sarah M. Baxter [1], Andrew JHL Snelling [2],
Daniel Tzu-Hsuan Chen [3], Emma C. Atakpa[3,4], Úna McMenamin[1],
Stuart A. McIntosh [5,6], Blánaid Hicks[1], Carol AC Coupland[2], Aaron J. Brady[7,8],
Finian J. Bannon[1] & Julia Hippisley-Cox[2]

The role of the gut microbiota in carcinogenesis is increasingly being acknowledged. Recent studies in multiple breast cancer mouse models have found that antibiotics, by altering the gut microbiota, can accelerate tumour growth. In humans, a recent cohort study restricted to triple negative breast cancer showed that breast cancer patients using a greater number of antibiotics had markedly worse survival. These studies have raised concerns about repeated antibiotic use in breast cancer patients. In this Registered Report, we investigated whether breast cancer patients using oral antibiotics had increased breast cancer-specific mortality. In population-based cohorts (n = 44,452), we did not observe a statistically significant association between antibiotic prescriptions after diagnosis and breast cancer-specific mortality (adjusted HR = 1.07 95% CI 0.87, 1.33) apart from prescriptions of 12 or more antibiotics (adjusted HR = 1.62 95% CI 1.31, 2.01). This association was weaker after adjustment for infections (adjusted HR = 1.44 95% 1.14, 1.81), when restricted to antibiotics within five years (adjusted HR = 1.33 95% 0.95, 1.84), and was similar for deaths from other causes (adjusted HR = 1.69 95% 1.19, 2.41). Frequent antibiotic users had higher cancer-specific mortality but the attenuation of associations in sensitivity analyses, and similar findings for other causes of death, suggest this increase may reflect residual confounding.
**Protocol registration:** The Stage 1 protocol for this Registered Report was accepted in principle on 7 November 2023. The protocol, as accepted by the journal, can be found at https://doi.org/10.6084/m9.figshare.24746721.v1.

## Antibiotic prescribing

In the UK, antibiotics are commonly prescribed, with around 50 antibiotic prescriptions issued per 100 people in England during 2017[1], and there is increasing evidence of unnecessary use[2]. In one UK study, half of antibiotic prescriptions did not have a clearly documented indication[3] (reference corrected from Stage 1 Registered Report), and

in another, a third of patients with respiratory tract infections received antibiotics despite national guidance recommending delaying or not using antibiotics[4]. Under the umbrella term of Antimicrobial Stewardship (AMS), the UK government has set targets to reduce inappropriate antibiotic prescribing[5] via strategies such as avoiding or delaying the prescribing of antibiotics in typically self-limiting

infections, using narrow-spectrum over broad-spectrum antibiotics, and minimising treatment duration. In addition, further AMS interventions such as training and decision support tools have been developed in primary care with the aim of reducing antibiotic prescribing[6].

Antibiotics are increasingly recognised as having a marked impact on the gut microbiota[7,8]. One review reported that antibiotics commonly used in primary care reduce both bacterial diversity and abundance of gut flora and that differences were related to the antibiotic class[8,9]. In addition, studies have shown that even short-term antibiotic use can result in altered gut microbiota for up to 2 years after treatment[9].

There is increasing evidence that the gut microbiota is important in cancer. Preclinical studies suggest the gut microbiota can alter cancer susceptibility and progression by several mechanisms, including influencing inflammation, inducing DNA damage, and via alteration of the immune system response[10]. Of particular importance to breast cancer, gut microbiota have been shown to influence oestrogen metabolism[11]. Additionally, studies have shown differences in microbiome composition in breast cancer patients compared with controls[12–15], such as one study that observed less diverse gut microbiota in breast cancer patients[13].

## Antibiotics and breast cancer

Recently, two preclinical studies suggested that antibiotics could impact breast cancer outcomes. In 2019[16], antibiotic use was shown to increase tumour dissemination in a hormone receptor-positive (luminal A) breast cancer mouse model. In 2021[17], this study was repeated, including the mouse model originally investigated (luminal A) but also other breast cancer mouse models (luminal B and the basal-like subtype). Concerningly, this study showed that antibiotic use (including commonly used antibiotics such as cephalexin) accelerated tumour growth in all breast cancer subtypes investigated. The researchers concluded that accelerated tumour growth likely resulted from the loss of beneficial microbiota, potentially via anti-tumorigenic species such as *Faecalibaculum rodentium*. These results suggest that antibiotic use may have a detrimental impact on breast cancer outcome by altering the gut microbiota. However, it is unclear whether the findings of these animal experiments will translate to humans[18].

In humans, epidemiological studies have shown increases in breast cancer risk of around 20% associated with antibiotic use[19] and weak evidence that breast cancer patients using antibiotics before diagnosis have less favourable tumour characteristics, such as higher grade[20]. However, there has been limited research into antibiotics used after diagnosis and survival or recurrence in breast cancer patients. An earlier cohort study of 4216 breast cancer patients[21] showed that frequent antibiotic users had a non-significant increase in the risk of second breast cancer events (including recurrences and second primary breast cancer) and more marked increases in second events with longer use of certain antibiotic classes. However, it did not investigate survival outcomes and, due to limited power, called for further research, particularly by antibiotic class. Another study observed worse survival in 120 patients with breast cancer who received antibiotics within 30 days of chemotherapy[22], but ignored antibiotic exposure after 30 days. A recent study[23] of 772 triple-negative breast cancer patients observed a significant increase in breast cancer-specific mortality rates of 5% for each antibiotic prescription used and 18% for each class of antibiotic used.

This evidence highlights the need for a large study of antibiotics in breast cancer patients, particularly given the high rates of antibiotic use and high prevalence of breast cancer (estimates suggest 1.2 million UK women will be living with breast cancer by 2030[24]). The primary objective was to investigate whether breast cancer patients who use or frequently use antibiotics have increased cancer-specific mortality. The study investigated cohorts of breast cancer patients in England and Wales identified from cancer registries (diagnosed 2000–2019) with linked national mortality data and primary care prescribing records.

## Results

A summary of the preregistered main analyses is shown in Table 1.

### Changes to data from Stage 1 report

The final analysis includes women diagnosed with breast cancer between 2000 and 2019 in both England and Wales, but originally, we had planned to include women diagnosed from 2000 to 2017.

### Patient inclusion

A flow-chart of breast cancer patients included in the main analysis is shown in Fig. 1. Overall, there were 28,750 stage 1–3 breast cancer patients in the England cohort and 15,702 stage 1–3 breast cancer patients in the Wales cohort included in the main analysis. There were 2495 breast cancer-specific deaths with a median follow-up of 5.7 years (interquartile range 3.4–8.9) in the England cohort, and there were 2082 breast cancer-specific deaths with a median follow-up of 7.3 (4.0–11.9) years in the Wales cohort.

### Patient characteristics

In England, 71% (20,459) of patients received an antibiotic after diagnosis, and 8% (2287) received 12 or more antibiotics. Similarly, in Wales, 79% (12,334) of patients received an antibiotic after diagnosis, and 15% (2331) received 12 or more antibiotics. Characteristics of patients by antibiotic use are shown in Tables 2 and 3. Breast cancer patients prescribed more antibiotics tended to be diagnosed in earlier years, reflecting their increased duration of follow-up. Stage was fairly similar across antibiotic categories. In England, 9% of antibiotic users had stage 3 disease compared with 9% of non-users. In Wales, 9% of users had stage 3 disease compared with 12% of non-users. Rates of surgery and radiotherapy were similar in antibiotic users and non-users in both cohorts. A slightly greater proportion of antibiotic users had chemotherapy compared with non-users (39% versus 35% in England; 39% versus 32% in Wales) and used tamoxifen in the year after diagnosis (40% versus 31% in England; 39% versus 28% in Wales). Other characteristics were generally similar in antibiotic users and non-users, including hormone receptor status (although there was considerable missing data), deprivation, other medication use, and co-morbidities.

**Question 1a\1b: Antibiotic use after cancer diagnosis and breast cancer-specific mortality.** The pooled associations between antibiotic use after diagnosis and cancer-specific mortality are shown in Table 4 and Fig. 2. Overall, there was no statistically significant difference in the rate of cancer-specific mortality in users of antibiotics after diagnosis (pooled adjusted HR = 1.07 95% CI 0.87,1.33) compared with non-users. However, there was heterogeneity between England and Wales ($P = 0.004$) with a weak association in England (adjusted HR = 1.19 95% CI 1.08, 1.31), but no association in Wales (adjusted HR = 0.96 95% CI 0.86, 1.07). There was evidence of a marked increase in breast cancer-specific mortality with use of 12 or more antibiotics (pooled adjusted HR = 1.62 95% CI 1.31, 2.01), which was consistent in England (adjusted HR = 1.81 95% CI 1.48, 2.20) and Wales (adjusted HR = 1.45, 95% CI 1.19, 1.78).

**Question 2: Antibiotic use by class after cancer diagnosis and cancer-specific mortality.** The associations between breast-cancer-specific mortality and antibiotic use by class after diagnosis are shown in Table 4 and Fig. 2. The associations by class of antibiotic appeared largely similar. For instance, excluding antibiotics with small numbers (clindamycin, sulfonamides, and other antibiotics), the pooled adjusted HR ranged from 1.09 (95% CI 0.87, 1.37) for penicillin to 1.28 (95% CI 1.11, 1.48) for metronidazole.

## Table 1 | Summary of main analyses and sensitivity analyses

| Question | Hypothesis | Outcome measure | Sampling plan (power analysis) | Analysis plan | Interpretation | Outcome |
|---|---|---|---|---|---|---|
| 1a. Main analysis (primary hypothesis): Do breast cancer patients who use antibiotics after diagnosis have increased cancer-specific mortality? | Breast cancer patients using antibiotics have a higher risk of breast cancer-specific mortality. | Cancer-specific mortality | All stage 1–3 breast cancer patients. Estimate over 95% power to detect HR of 1.2 in antibiotic users compared with antibiotic non-users. | Cox regression calculating the association between any antibiotic use and cancer-specific mortality, after adjusting for confounders[a]. | HR for any antibiotic use >1.2 (and $p < 0.05$) provides support for increased cancer-specific mortality in antibiotic users—but further analysis of exposure-response and sensitivity analyses is necessary. HR for any antibiotic use $p \geq 0.05$ indicates no statistically significant association. | 1a. Hypothesis not confirmed: Risk of breast cancer-specific mortality not higher in antibiotic users. |
| 1b. Main analysis: Do breast cancer patients who frequently use antibiotics (specifically 12 or more prescriptions) after diagnosis have increased cancer-specific mortality? | Breast cancer patients using more antibiotics have a higher risk of breast cancer-specific mortality. | Cancer-specific mortality | All stage 1–3 breast cancer patients. Estimate over 95% power to detect HR of 1.25 in users of 12 or more antibiotics compared with users of fewer than 12. | Cox regression calculating the association between antibiotic use in categories (none, 1–5, 6–11, 12 or more) and cancer-specific mortality, after adjusting for confounders[a]. | HR for 12 or more antibiotics > 1.25 (and $p < 0.05$) provides support for increased cancer-specific mortality with frequent antibiotic use—sensitivity analyses necessary. HR for 12 or more antibiotics $p \geq 0.05$ indicates no statistically significant association. | 1b. Hypothesis confirmed: Risk of breast cancer-specific mortality is higher in users of 12 or more antibiotics. |
| 2. Main analysis: Do breast cancer patients who use a specific antibiotic class after diagnosis have increased cancer-specific mortality, particularly cephalosporins? | Breast cancer patients using cephalosporins (or other specific antibiotic classes) have a higher risk of breast cancer-specific mortality. | Cancer-specific mortality | All stage 1–3 breast cancer patients. Estimate over 95% power to detect HR of 1.2 in users of cephalosporins compared with antibiotic non-users. | Cox regression calculating the association between antibiotic use by class and cancer-specific mortality, after adjusting for confounders[a]. | HR for cephalosporins (or other specific antibiotic class) > 1.2 (and $p < 0.05$) provides support for increased cancer-specific mortality in that antibiotic class—but further analysis of exposure-response and sensitivity analyses is necessary. HR for all antibiotic classes $p \geq 0.05$ indicates no statistically significant association. | 2. Hypothesis not confirmed: Risk of breast cancer-specific mortality not higher in cephalosporin users or users of other antibiotics apart from clindamycin, metronidazole, and nitrofurantoin. |
| 3. Sensitivity analysis (for analyses 1a and 1b): How consistent is the association between antibiotic use after diagnosis and cancer-specific mortality across different analyses? The association will be determined after the following changes in the analysis: (a) Ignoring antibiotic use before diagnosis; (b) Additionally, adjusting for prior antibiotic use; (c) Comparing broad-spectrum to narrow-spectrum antibiotics; (d) Additional adjusting for infections; (e) Further analyses to reduce reverse causation; (f) Restricted to oestrogen receptor-positive breast cancer; (g) Additionally, adjusting for smoking and BMI; (h) Changing the age range; (i) Changing the outcome to breast cancer-specific death (based upon breast cancer as any cause); (l) Changing the outcome to all-cause mortality. | Breast cancer patients using antibiotics or frequently using antibiotics have a higher risk of breast cancer-specific mortality. | Cancer-specific mortality (except where otherwise stated) | All stage 1–3 breast cancer patients, but the study population will vary in different sensitivity analyses. In many analyses, the study would have over 95% power to detect an HR of 1.2 in antibiotic users compared with antibiotic non-users and an HR of 1.25 for frequent antibiotic use, but power will vary across analyses. | Cox regression calculating the association between any and frequent antibiotic use and cancer-specific mortality, after adjusting for confounders[a] and altering the analyses as described. | Should the sensitivity analyses consistently show an HR for any antibiotic use >1.2 or an HR for frequent antibiotic use >1.25, this would provide support for increased cancer-specific mortality in antibiotic users. Should 2 or more sensitivity analyses have HRs close to 1, this would provide evidence against a causal association between antibiotic use and cancer-specific mortality. | 3a. Hypothesis not confirmed: Risk of breast cancer-specific mortality consistently not higher in antibiotic users across sensitivity analyses. 3b. Hypothesis confirmed: Risk of breast cancer-specific mortality consistently higher in users of 12 or more antibiotics across sensitivity analyses. |

[a]Except where otherwise stated, the adjusted model contains the following covariates: age at diagnosis, year of diagnosis, stage, grade, surgery, radiotherapy, chemotherapy, hormone therapy use (after diagnosis), hormone replacement therapy use (before diagnosis), other medication use (after diagnosis, including statin, aspirin and metformin) and deprivation (in fifths).

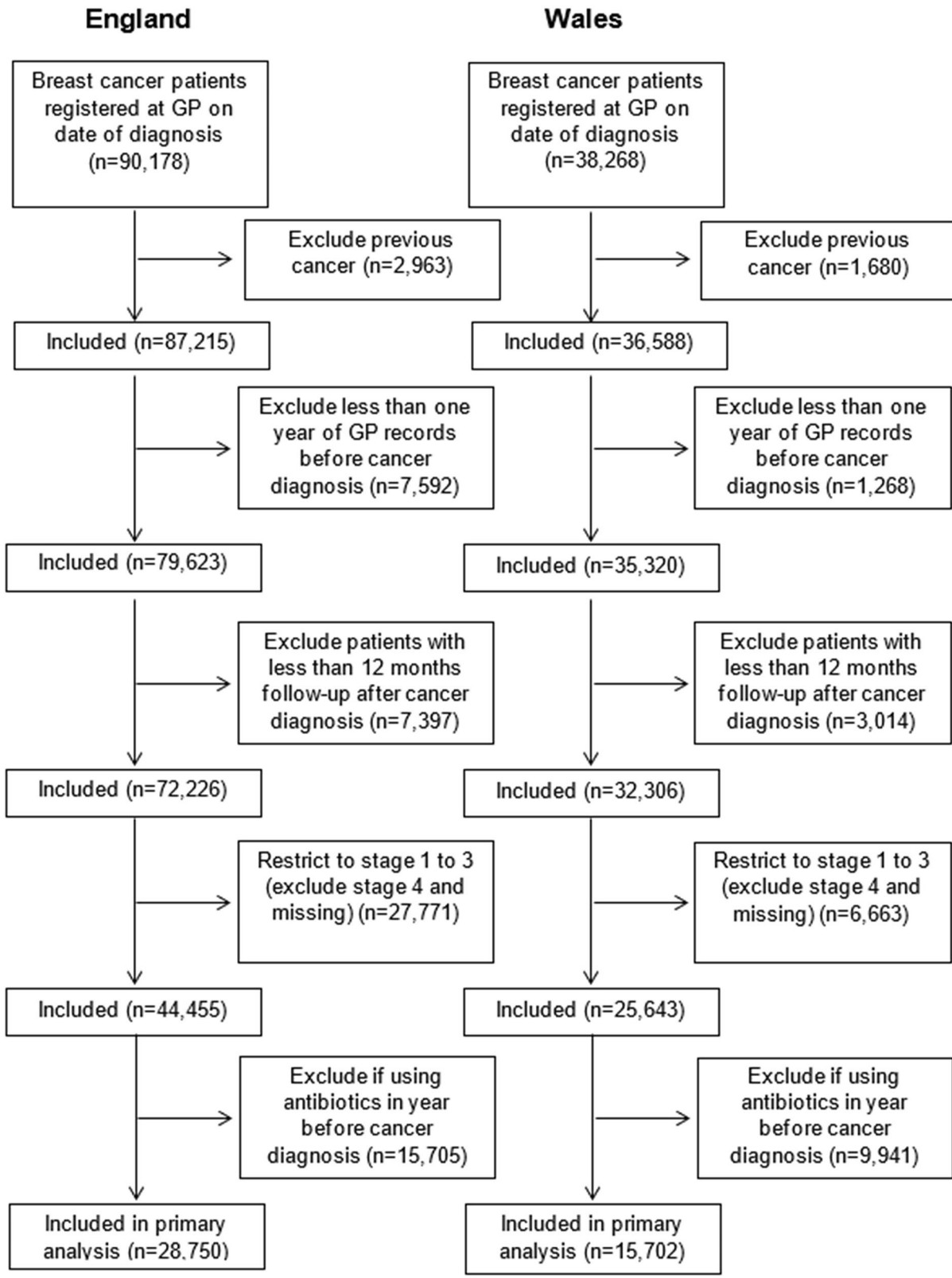

**Fig. 1 | Flow chart of patients included in the study in England and Wales.**

**Question 3: Sensitivity analysis of antibiotic use after cancer diagnosis and cancer-specific mortality.** Sensitivity analyses are shown in Table 5 and Supplementary Table 1, including subgroup analyses shown in Table 6 and Supplementary Table 2. The findings of most analyses were similar to the main analysis, but the associations were somewhat attenuated after adjusting for infections, when using longer lag periods, and when restricting to antibiotics prescribed in the first 5 years after diagnosis. For instance, after additionally adjusting for infections, the pooled adjusted hazard ratio in antibiotic users was 1.02 (95% CI 0.82, 1.27) and in users of 12 or more antibiotics was 1.44 (95% CI 1.14, 1.81) compared with non-users. When using a 2-year lag, compared with non-users, the pooled adjusted hazard ratio in users was

**Table 2 | Characteristics of breast cancer patients by antibiotic use after diagnosis in England and Wales 2000–2019**

| Characteristics | England | | | | | Wales | | | | |
|---|---|---|---|---|---|---|---|---|---|---|
| | None | 1+ | 1–5 | 6–11 | 12+ | None | 1+ | 1–5 | 6–11 | 12+ |
| Age: 18–39 | 348 (4%) | 801 (4%) | 576 (4%) | 146 (4%) | 79 (3%) | 106 (3%) | 490 (4%) | 276 (4%) | 131 (5%) | 83 (4%) |
| 40–49 | 1220 (15%) | 3133 (15%) | 2252 (16%) | 565 (15%) | 316 (14%) | 389 (12%) | 1935 (16%) | 1152 (16%) | 415 (15%) | 368 (16%) |
| 50–59 | 2179 (26%) | 5647 (28%) | 4011 (28%) | 1031 (27%) | 605 (26%) | 865 (26%) | 3455 (28%) | 2025 (28%) | 779 (28%) | 651 (28%) |
| 60–69 | 2284 (28%) | 5600 (27%) | 3838 (27%) | 1058 (27%) | 704 (31%) | 958 (28%) | 3550 (29%) | 2062 (28%) | 795 (29%) | 693 (30%) |
| 70–79 | 1431 (17%) | 3541 (17%) | 2376 (17%) | 732 (19%) | 433 (19%) | 618 (18%) | 2026 (16%) | 1177 (16%) | 431 (16%) | 418 (18%) |
| 80+ | 829 (10%) | 1737 (8%) | 1255 (9%) | 332 (9%) | 150 (7%) | 432 (13%) | 878 (7%) | 574 (8%) | 186 (7%) | 118 (5%) |
| Year: 2000–2004 | 475 (6%) | 2540 (12%) | 1250 (9%) | 630 (16%) | 660 (29%) | 589 (17%) | 2582 (21%) | 1094 (15%) | 664 (24%) | 824 (35%) |
| 2005–2009 | 510 (6%) | 2710 (13%) | 1467 (10%) | 674 (17%) | 569 (25%) | 426 (13%) | 3004 (24%) | 1433 (20%) | 769 (28%) | 802 (34%) |
| 2010–2014 | 1828 (22%) | 6238 (30%) | 4144 (29%) | 1391 (36%) | 703 (31%) | 789 (23%) | 3696 (30%) | 2289 (32%) | 858 (31%) | 549 (24%) |
| 2015–2019 | 5478 (66%) | 8971 (44%) | 7447 (52%) | 1169 (30%) | 355 (16%) | 1564 (46%) | 3052 (25%) | 2450 (34%) | 446 (16%) | 156 (7%) |
| Stage: 1 | 4057 (49%) | 9897 (48%) | 6810 (48%) | 1936 (50%) | 1151 (50%) | 1596 (47%) | 6056 (49%) | 3455 (48%) | 1391 (51%) | 1210 (52%) |
| 2 | 3494 (42%) | 8821 (43%) | 6187 (43%) | 1640 (42%) | 994 (43%) | 1377 (41%) | 5122 (42%) | 3068 (42%) | 1117 (41%) | 937 (40%) |
| 3 | 740 (9%) | 1741 (9%) | 1311 (9%) | 288 (7%) | 142 (6%) | 395 (12%) | 1156 (9%) | 743 (10%) | 229 (8%) | 184 (8%) |
| Grade: 1 | 1373 (17%) | 3431 (17%) | 2322 (16%) | 677 (18%) | 432 (19%) | 482 (14%) | 1888 (15%) | 1080 (15%) | 399 (15%) | 409 (18%) |
| 2 | 4285 (52%) | 10,372 (51%) | 7331 (51%) | 1928 (50%) | 1113 (49%) | 1544 (46%) | 5951 (48%) | 3471 (48%) | 1336 (49%) | 1144 (49%) |
| 3 | 2410 (29%) | 6161 (30%) | 4325 (30%) | 1161 (30%) | 675 (30%) | 980 (29%) | 3378 (27%) | 2042 (28%) | 760 (28%) | 576 (25%) |
| Missing | 223 (3%) | 495 (2%) | 330 (2%) | 98 (3%) | 67 (3%) | 362 (11%) | 1117 (9%) | 673 (9%) | 242 (9%) | 202 (9%) |
| Surgery | 7782 (94%) | 19,714 (96%) | 13,760 (96%) | 3735 (97%) | 2219 (97%) | 3058 (91%) | 11,752 (95%) | 6895 (95%) | 2630 (96%) | 2227 (96%) |
| Radiotherapy | 5827 (70%) | 14,684 (72%) | 10,359 (72%) | 2761 (71%) | 1564 (68%) | 2071 (61%) | 7752 (63%) | 4706 (65%) | 1702 (62%) | 1344 (58%) |
| Chemotherapy | 2904 (35%) | 8059 (39%) | 5744 (40%) | 1480 (38%) | 835 (37%) | 1081 (32%) | 4833 (39%) | 2840 (39%) | 1110 (41%) | 883 (38%) |
| AIs[a] | 4043 (49%) | 9228 (45%) | 6686 (47%) | 1740 (45%) | 802 (35%) | 1730 (51%) | 5713 (46%) | 3643 (50%) | 1194 (44%) | 876 (38%) |
| Tamoxifen[b] | 2543 (31%) | 8175 (40%) | 5345 (37%) | 1693 (44%) | 1137 (50%) | 935 (28%) | 4780 (39%) | 2509 (35%) | 1157 (42%) | 1114 (48%) |
| ER: Negative | 823 (10%) | 1682 (8%) | 1257 (9%) | 273 (7%) | 152 (7%) | 361 (11%) | 1227 (10%) | 781 (11%) | 264 (10%) | 182 (8%) |
| Positive | 4594 (55%) | 10,693 (52%) | 7825 (55%) | 1949 (50%) | 919 (40%) | 2220 (66%) | 7770 (63%) | 4903 (67%) | 1641 (60%) | 1226 (53%) |
| Missing | 2874 (35%) | 8084 (40%) | 5226 (37%) | 1642 (42%) | 1216 (53%) | 787 (23%) | 3337 (27%) | 1582 (22%) | 832 (30%) | 923 (40%) |
| PR: Negative | 1068 (13%) | 2191 (11%) | 1656 (12%) | 366 (9%) | 169 (7%) | 485 (14%) | 1666 (14%) | 1080 (15%) | 348 (13%) | 238 (10%) |
| Positive | 2490 (30%) | 5404 (26%) | 4022 (28%) | 983 (25%) | 399 (17%) | 1056 (31%) | 3363 (27%) | 2208 (30%) | 686 (25%) | 469 (20%) |
| Missing | 4733 (57%) | 12,864 (63%) | 8630 (60%) | 2515 (65%) | 1719 (75%) | 1827 (54%) | 7305 (59%) | 3978 (55%) | 1703 (62%) | 1624 (70%) |
| HER2: Negative | 4877 (59%) | 10,409 (51%) | 7884 (55%) | 1759 (46%) | 766 (33%) | 2105 (62%) | 6680 (54%) | 4352 (60%) | 1381 (50%) | 947 (41%) |
| Positive | 661 (8%) | 1758 (9%) | 1286 (9%) | 318 (8%) | 154 (7%) | 289 (9%) | 1211 (10%) | 766 (11%) | 252 (9%) | 193 (8%) |
| Missing | 2753 (33%) | 8292 (41%) | 5138 (36%) | 1787 (46%) | 1367 (60%) | 974 (29%) | 4443 (36%) | 2148 (30%) | 1104 (40%) | 1191 (51%) |

[a]Aromatase inhibitors in the first year after diagnosis.
[b]Tamoxifen in the first year after diagnosis

1.04 (95% CI 0.91, 1.20), and in users of 12 or more antibiotics was 1.40 (95% CI 1.19, 1.65) compared with non-users, and there was no heterogeneity between countries. Restricting to antibiotics used in the first 5 years after diagnosis, the pooled adjusted hazard ratio in users was 1.11 (95% CI 0.94, 1.30) and in users of 12 or more antibiotics was 1.33 (95% CI 0.95, 1.84) compared with non-users, and there was no heterogeneity between countries. The associations appeared stronger when analyses focused on antibiotics used later in life. For instance, after removing antibiotics in the 12 months after diagnoses the adjusted hazard ratio in users was 1.28 (95% CI 1.18, 1.38) and in users of 12 or more antibiotics was 1.85 (95% CI 1.58, 2.17) compared with non-users and there was no heterogeneity between countries The association between antibiotics and breast cancer specific mortality appeared similar in oestrogen receptor positive (pooled adjusted HR for any use 1.08 95% CI 0.96, 1.22 and 12 or more prescriptions 1.48 95% CI 1.06, 2.06) and oestrogen receptor negative breast cancer (pooled adjusted HR for any use 0.89 95% CI 0.69, 1.14 and 12 or more prescriptions 1.95 95% CI 1.27, 2.98).

Table 7 shows the analysis of different mortality outcomes. The association with antibiotic use after diagnosis appeared stronger when investigating all-cause mortality. Compared with non-users, the pooled adjusted hazard ratio for all-cause mortality in users was 1.11 (95% CI 0.87, 1.42), and in users of 12 or more antibiotics was 1.76 (95% CI 1.36, 2.29).

**Results of unplanned analyses (Stage 2 analyses): Additional analysis of antibiotic use after diagnosis and mortality.** Analysis of additional outcomes is also shown in Table 7. There was a strong association between antibiotic use and deaths where breast cancer was not the underlying cause, and also for deaths where breast cancer was not mentioned on the death certificate. The pooled adjusted hazard ratio for deaths where breast cancer was not the underlying cause in users of 12 or more antibiotics was 1.69 (95% CI 1.19, 2.41) compared with non-users.

## Discussion

In this Registered Report, prescriptions for antibiotics after breast cancer diagnosis were not significantly associated with breast cancer-specific mortality, apart from prescriptions for 12 or more antibiotics, which were associated with a pooled 62% increased risk of breast cancer-specific mortality. This increased risk in patients prescribed 12 or more antibiotics was reduced in sensitivity analyses adjusting for infections to 44%, and also reduced in sensitivity analyses limiting reverse causality (by applying longer lags or restricting to antibiotics in

**Table 3 | Further characteristics of breast cancer patients by antibiotic use after diagnosis in England and Wales**

| Characteristics | England | | | | | Wales | | | | |
|---|---|---|---|---|---|---|---|---|---|---|
| | None | 1+ | 1–5 | 6–11 | 12+ | None | 1+ | 1–5 | 6–11 | 12+ |
| Deprivation | | | | | | | | | | |
| 1st fifth (deprived) | 1069 (13%) | 2802 (14%) | 1924 (13%) | 532 (14%) | 346 (15%) | 474 (14%) | 1798 (15%) | 996 (14%) | 397 (15%) | 405 (17%) |
| 2nd fifth | 1469 (18%) | 3549 (17%) | 2480 (17%) | 672 (17%) | 397 (17%) | 644 (19%) | 2257 (18%) | 1279 (18%) | 518 (19%) | 460 (20%) |
| 3rd fifth | 1664 (20%) | 4153 (20%) | 2901 (20%) | 790 (20%) | 462 (20%) | 653 (19%) | 2625 (21%) | 1524 (21%) | 576 (21%) | 525 (23%) |
| 4th fifth | 1925 (23%) | 4796 (23%) | 3347 (23%) | 908 (23%) | 541 (24%) | 714 (21%) | 2585 (21%) | 1599 (22%) | 567 (21%) | 419 (18%) |
| 5th fifth (affluent) | 2164 (26%) | 5159 (25%) | 3656 (26%) | 962 (25%) | 541 (24%) | 878-83[5] (26%) | 3062[5] (25%) | 1863-8[5] (26%) | 674-9[5] (25%) | 517-22[5] (22%) |
| Myocardial infarction | 102 (1%) | 295 (1%) | 176 (1%) | 80 (2%) | 39 (2%) | 67 (2%) | 156 (1%) | 93 (1%) | 29 (1%) | 34 (1%) |
| Heart failure | 88 (1%) | 236 (1%) | 159 (1%) | 53 (1%) | 24 (1%) | 96 (3%) | 206 (2%) | 117 (2%) | 49 (2%) | 40 (2%) |
| PVD[a] | 58 (1%) | 176 (1%) | 104 (1%) | 45 (1%) | 27 (1%) | 58 (2%) | 176 (1%) | 97 (1%) | 37 (1%) | 42 (2%) |
| Stroke | 221 (3%) | 634 (3%) | 404 (3%) | 130 (3%) | 100 (4%) | 139 (4%) | 420 (3%) | 240 (3%) | 95 (3%) | 85 (4%) |
| COPD[b] | 75 (1%) | 345 (2%) | 188 (1%) | 76 (2%) | 81 (4%) | 127 (4%) | 543 (4%) | 275 (4%) | 122 (4%) | 146 (6%) |
| Hemiplegia | 23 (0%) | 66 (0%) | 43 (0%) | 11 (0%) | 12 (1%) | 24 (1%) | 92 (1%) | 55 (1%) | 14 (1%) | 23 (1%) |
| Dementia | 60 (1%) | 104 (1%) | 71 (0%) | 21 (1%) | 12 (1%) | 40 (1%) | 81 (1%) | 52 (1%) | 18 (1%) | 11 (0%) |
| Liver diseases | 86 (1%) | 226 (1%) | 157 (1%) | 42 (1%) | 27 (1%) | 27 (1%) | 101 (1%) | 60 (1%) | 20 (1%) | 21 (1%) |
| Peptic ulcer | 127 (2%) | 421 (2%) | 263 (2%) | 96 (2%) | 62 (3%) | 41 (1%) | 211 (2%) | 102 (1%) | 48 (2%) | 61 (3%) |
| Diabetes | 529 (6%) | 1462 (7%) | 977 (7%) | 305 (8%) | 180 (8%) | 381 (11%) | 1405 (11%) | 828 (11%) | 315 (12%) | 262 (11%) |
| Chronic kidney disease | 481 (6%) | 1234 (6%) | 838 (6%) | 274 (7%) | 122 (5%) | 229 (7%) | 669 (5%) | 402 (6%) | 155 (6%) | 112 (5%) |
| Statin (after)[3] | 1480 (18%) | 4038 (20%) | 2715 (19%) | 838 (22%) | 485 (21%) | 566 (17%) | 2431 (20%) | 1376 (19%) | 557 (20%) | 498 (21%) |
| Aspirin (after)[3] | 479 (6%) | 1788 (9%) | 1086 (8%) | 415 (11%) | 287 (13%) | 278 (8%) | 1319 (11%) | 687 (9%) | 312 (11%) | 320 (14%) |
| Metformin (after)[3] | 351 (4%) | 1004 (5%) | 658 (5%) | 215 (6%) | 131 (6%) | 137 (4%) | 504 (4%) | 285 (4%) | 111 (4%) | 108 (5%) |
| HRT (before)[4] | 1654 (20%) | 5217 (25%) | 3294 (23%) | 1102 (29%) | 821 (36%) | 631 (19%) | 3198 (26%) | 1697 (23%) | 757 (28%) | 744 (32%) |
| Smoking: Current | 498 (6%) | 1375 (7%) | 910 (6%) | 295 (8%) | 170 (7%) | 413 (12%) | 1977 (16%) | 1108 (15%) | 438 (16%) | 431 (18%) |
| Past | 1710 (21%) | 4819 (24%) | 3395 (24%) | 896 (23%) | 528 (23%) | 570 (17%) | 2470 (20%) | 1508 (21%) | 534 (20%) | 428 (18%) |
| Never | 5073 (61%) | 11,717 (57%) | 8329 (58%) | 2159 (56%) | 1229 (54%) | 1863 (55%) | 6850 (56%) | 4086 (56%) | 1515 (55%) | 1249 (54%) |
| Missing | 560 (7%) | 1164 (6%) | 744 (5%) | 234 (6%) | 186 (8%) | 522 (15%) | 1037 (8%) | 564 (8%) | 250 (9%) | 223 (10%) |
| BMI: <18.5 | 137 (2%) | 219 (1%) | 163 (1%) | 36 (1%) | 20 (1%) | 38 (1%) | 121 (1%) | 78 (1%) | 24 (1%) | 19 (1%) |
| 18.5–24.9 | 2660 (32%) | 5949 (29%) | 4330 (30%) | 1076 (28%) | 543 (24%) | 917 (27%) | 3414 (28%) | 2120 (29%) | 732 (27%) | 562 (24%) |
| 25–29.9 | 2299 (28%) | 5855 (29%) | 4067 (28%) | 1138 (29%) | 650 (28%) | 822 (24%) | 3454 (28%) | 2022 (28%) | 769 (28%) | 663 (28%) |
| ≥30 | 1636 (20%) | 5014 (25%) | 3326 (23%) | 1016 (26%) | 672 (29%) | 669 (20%) | 3002 (24%) | 1655 (23%) | 695 (25%) | 652 (28%) |
| Missing | 1559 (19%) | 3422 (17%) | 2422 (17%) | 598 (15%) | 402 (18%) | 922 (27%) | 2343 (19%) | 1391 (19%) | 517 (19%) | 435 (19%) |
| Mean (s.d.) | 27.0 (6.0) | 27.7 (5.8) | 27.5 (5.7) | 28.0 (5.9) | 28.7 (6.2) | 27.2 (5.5) | 27.8 (5.8) | 27.5 (5.6) | 28.0 (5.7) | 28.6 (6.2) |

[a]Peripheral vascular disease.
[b]Chronic obstructive pulmonary disease. [3]Medication use in the year after diagnosis. [4]Hormone replacement therapy use before diagnosis. [5]Range used to preserve disclosure control.

the 5 years after diagnosis). Additionally, in patients prescribed 12 or more antibiotics, there was a similar increased risk of 69% in deaths for which breast cancer was not the underlying cause.

**Comparison with previous studies**

Our findings for antibiotic use after diagnosis are broadly consistent with previous studies, but these were based upon much smaller numbers. An earlier cohort study of 4216 breast cancer patients[21] observed an increased risk in breast cancer recurrence associated with frequent (defined as four prescriptions in a year) antibiotic use after diagnosis of 38% before adjustment for confounders (HR = 1.38 95% CI 1.03, 1.84), which reduced to 13% (adjusted HR = 1.13 95% CI 0.83, 1.53) after adjustment for confounders including stage, treatment and some infections. A recent study[23] of 772 triple-negative breast cancer patients did not observe a significant association with any antibiotic use after diagnosis (HR = 1.39 95% CI 0.93, 2.29), which could reflect the study size, but observed an increase in breast cancer-specific mortality with increasing numbers of antibiotic prescriptions (HR per prescription = 1.05 95% CI 1.01, 1.08). Another study[25], from the UK, observed a 36% (adjusted HR = 1.36 95% CI 1.23, 1.49) increased risk of all-cause mortality in antibiotic users compared with non-users, but it is more difficult to directly compare with our findings because it investigated antibiotic use in the 3-month period before breast cancer diagnosis rather than after diagnosis.

**Interpretation**

The cause of the increased breast cancer-specific mortality in patients prescribed 12 or more antibiotics after diagnosis is unknown. This association is consistent with animal studies that suggest that repeated antibiotic use could have a detrimental carcinogenic impact on the microbiome, reducing beneficial microbiota, and resulting in increased risk of breast cancer recurrence and death[16,17]. However, this association could reflect confounding by indication. Patients who were prescribed 12 or more antibiotics are likely to have had multiple infections, and the association was partly but not completely attenuated in sensitivity analysis, additionally adjusting for recorded GP and hospital infections after diagnosis, but residual confounding by infection remains likely, as these medical records will not capture all infections. A previous study of a breast cancer cohort showed a 37% increase in breast cancer-specific mortality in patients who were hospitalised with an infection[26], and recent evidence from animal models and observational studies suggests that respiratory infections can awaken dormant breast cancer cells, increasing the risk of cancer recurrence and mortality[27]. Breast cancer patients receiving multiple antibiotics may have underlying health conditions, which could reduce their likelihood of receiving comprehensive cancer treatment and lead to worse breast cancer outcomes. Confounding by cancer progression, a recognised problem in studies of cancer survival, is also possible[28]. In our study, we did not have data on cancer recurrence. Patients whose cancer

**Table 4 | Analysis of antibiotics after diagnosis and breast cancer-specific mortality in England and Wales and pooled**

| Exposure | England | | | | Wales | | | | P for hetero[b] | Pooled adjusted[a] HR (95% CI) | P |
|---|---|---|---|---|---|---|---|---|---|---|---|
| | P-years | Events | Unadjusted HR (95% CI) | Adjusted[a] HR (95% CI) | P-years | Events | Unadjusted HR (95% CI) | Adjusted[a] HR (95% CI) | | | |
| **Antibiotics** | | | | | | | | | | | |
| None | 69,935 | 822 | 1.00 (ref. cat.) | 1.00 (ref. cat.) | 36,194 | 692 | 1.00 (ref. cat.) | 1.00 (ref. cat.) | | 1.00 (ref. cat.) | |
| 1+ prescriptions | 123,282 | 1673 | 1.45 (1.33, 1.59) | 1.19 (1.08, 1.31) | 94,145 | 1390 | 1.01 (0.91, 1.12) | 0.96 (0.86, 1.07) | 0.004 | 1.07 (0.87, 1.33) | 0.521 |
| 1–5 prescriptions | 92,692 | 1243 | 1.38 (1.26, 1.52) | 1.15 (1.05, 1.27) | 63,076 | 939 | 0.95 (0.86, 1.06) | 0.93 (0.83, 1.04) | 0.005 | 1.04 (0.84, 1.28) | 0.749 |
| 6–11 prescriptions | 20,191 | 276 | 1.86 (1.60, 2.17) | 1.39 (1.19, 1.62) | 18,843 | 258 | 1.19 (1.01, 1.39) | 1.05 (0.88, 1.25) | 0.018 | 1.21 (0.92, 1.59) | 0.173 |
| 12+ prescriptions | 10,399 | 154 | 2.43 (2.00, 2.95) | 1.81 (1.48, 2.20) | 12,226 | 193 | 1.70 (1.41, 2.05) | 1.45 (1.19, 1.78) | 0.127 | 1.62 (1.31, 2.01) | <0.001 |
| Penicillin | 96,532 | 1341 | 1.52 (1.38, 1.67) | 1.22 (1.11, 1.35) | 76,618 | 1123 | 1.02 (0.92, 1.14) | 0.97 (0.86, 1.09) | 0.002 | 1.09 (0.87, 1.37) | 0.464 |
| Cephalosporin | 13,347 | 200 | 1.88 (1.60, 2.22) | 1.31 (1.10, 1.55) | 15,325 | 244 | 1.28 (1.10, 1.50) | 1.06 (0.89, 1.25) | 0.084 | 1.18 (0.96, 1.45) | 0.126 |
| Tetracyclines | 20,324 | 255 | 1.51 (1.30, 1.76) | 1.28 (1.10, 1.50) | 17,990 | 237 | 1.06 (0.90, 1.24) | 1.03 (0.86, 1.23) | 0.066 | 1.15 (0.93, 1.43) | 0.198 |
| Macrolides | 27,063 | 362 | 1.55 (1.36, 1.77) | 1.26 (1.10, 1.45) | 25,026 | 373 | 1.14 (1.00, 1.31) | 1.04 (0.89, 1.21) | 0.059 | 1.15 (0.95, 1.39) | 0.158 |
| Clindamycin | 1141 | 22 | 2.15 (1.40, 3.29) | 1.69 (1.10, 2.59) | 958 | 18 | 1.43 (0.89, 2.29) | 1.22 (0.76, 1.96) | 0.319 | 1.46 (1.06, 2.01) | 0.020 |
| Sulphonamides | 163 | 0–5[5] | 3.44 (1.42, 8.29) | 2.39 (0.99, 5.77) | 225 | 0–5[5] | 0.72 (0.18, 2.88) | 1.02 (0.25, 4.12) | 0.315 | 1.87 (0.88, 3.96) | 0.103 |
| Trimethoprim | 31,057 | 395 | 1.54 (1.35, 1.76) | 1.20 (1.05, 1.37) | 30,542 | 425 | 1.07 (0.94, 1.23) | 1.03 (0.89, 1.19) | 0.128 | 1.11 (0.95, 1.29) | 0.177 |
| Metronidazole[3] | 9187 | 123 | 1.63 (1.34, 1.99) | 1.33 (1.09, 1.63) | 8542 | 141 | 1.30 (1.08, 1.58) | 1.24 (1.01, 1.51) | 0.601 | 1.28 (1.11, 1.48) | <0.001 |
| Quinolones | 8181 | 129 | 1.97 (1.62, 2.39) | 1.31 (1.07, 1.60) | 8219 | 137 | 1.31 (1.08, 1.59) | 1.08 (0.88, 1.32) | 0.179 | 1.19 (0.98, 1.44) | 0.075 |
| Nitrofurantoin[4] | 24,685 | 299 | 1.39 (1.21, 1.60) | 1.20 (1.04, 1.39) | 13,628 | 192 | 1.11 (0.93, 1.31) | 1.23 (1.02, 1.49) | 0.806 | 1.21 (1.08, 1.36) | 0.001 |
| Other antibiotics | 512 | 10 | 2.27 (1.21, 4.24) | 1.57 (0.84, 2.94) | 237 | 6 | 1.90 (0.85, 4.25) | 1.50 (0.62, 3.62) | 0.930 | 1.55 (0.93, 2.58) | 0.095 |

[a]Adjusted model contains age at diagnosis, year of diagnosis, deprivation, stage, grade, surgery, radiotherapy, chemotherapy, tamoxifen use (in year after diagnosis), aromatase inhibitor use (in year after diagnosis), Charlson comorbidities (before diagnosis), hormone replacement therapy (before diagnosis) and statin, aspirin, and metformin use (after diagnosis) and is based upon a complete case analysis.
[b]P-value for heterogeneity comparing adjusted hazard ratios for England and Wales. [3]Metronidazole and tinidazole. [4]Nitrofurantoin and methenamine. [5]Range used to preserve disclosure control.

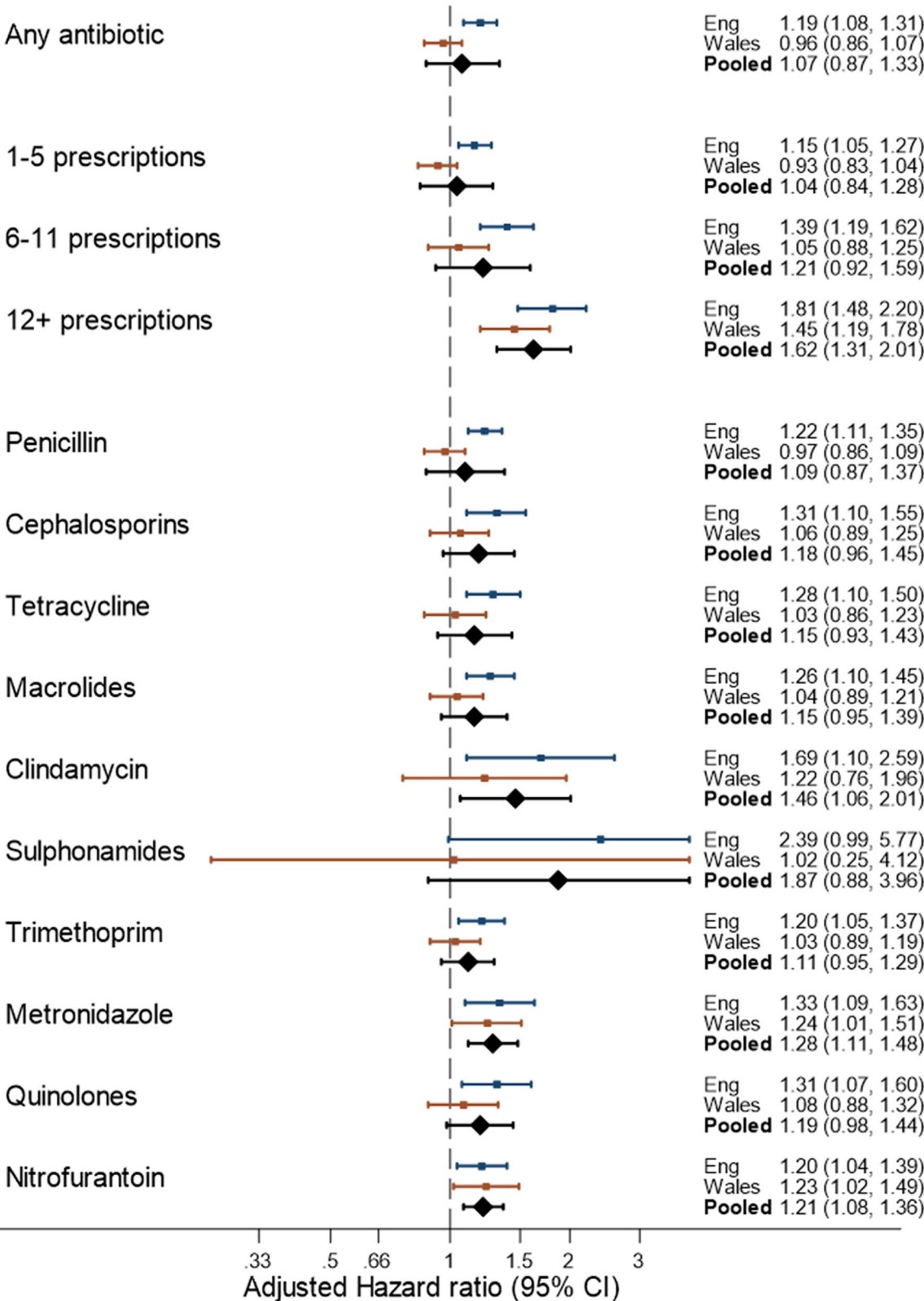

**Fig. 2 | Adjusted hazard ratios for the association between antibiotic use and breast cancer-specific mortality, by country and pooled.** The figure shows Hazard Ratios (and 95% Confidence Intervals) for antibiotics after diagnosis after adjustment for age at diagnosis, year of diagnosis, deprivation, stage, grade, surgery, radiotherapy, chemotherapy, tamoxifen use (in year after diagnosis), aromatase inhibitor use (in year after diagnosis), Charlson comorbidities (before diagnosis), hormone replacement therapy (before diagnosis) and statin, aspirin, and metformin use (after diagnosis).

**Table 5 | Sensitivity analyses for the association between antibiotics and breast cancer-specific mortality**

| ABs | England | | Wales | | P for hetero[b] | Pooled adjusted[a] HR (95% CI) | P |
|---|---|---|---|---|---|---|---|
| | Unadjusted HR (95% CI) | Adjusted[a] HR (95% CI) | Unadjusted HR (95% CI) | Adjusted[a] HR (95% CI) | | | |
| Main analysis (events = 4577, person-years = 323,556) | | | | | | | |
| Any | 1.45 (1.33, 1.59) | 1.19 (1.08, 1.31) | 1.01 (0.91, 1.12) | 0.96 (0.86, 1.07) | 0.004 | 1.07 (0.87, 1.33) | 0.521 |
| 1-5 | 1.38 (1.26, 1.52) | 1.15 (1.05, 1.27) | 0.95 (0.86, 1.06) | 0.93 (0.83, 1.04) | 0.005 | 1.04 (0.84, 1.28) | 0.749 |
| 6-11 | 1.86 (1.60, 2.17) | 1.39 (1.19, 1.62) | 1.19 (1.01, 1.39) | 1.05 (0.88, 1.25) | 0.018 | 1.21 (0.92, 1.59) | 0.173 |
| 12+ | 2.43 (2.00, 2.95) | 1.81 (1.48, 2.20) | 1.70 (1.41, 2.05) | 1.45 (1.19, 1.78) | 0.127 | 1.62 (1.31, 2.01) | <0.001 |
| Adjusting for antibiotic use in the year before diagnosis[3] (events = 7484, person-years = 503,230) | | | | | | | |
| Any | 1.47 (1.37, 1.59) | 1.21 (1.12, 1.30) | 1.08 (0.99, 1.17) | 1.02 (0.93, 1.12) | 0.008 | 1.11 (0.95, 1.31) | 0.197 |
| 1-5 | 1.38 (1.27, 1.49) | 1.17 (1.08, 1.26) | 1.01 (0.93, 1.11) | 0.99 (0.90, 1.09) | 0.012 | 1.08 (0.92, 1.26) | 0.357 |
| 6-11 | 1.81 (1.62, 2.02) | 1.38 (1.23, 1.55) | 1.21 (1.07, 1.37) | 1.12 (0.98, 1.28) | 0.021 | 1.25 (1.02, 1.53) | 0.034 |
| 12+ | 2.14 (1.89, 2.42) | 1.69 (1.48, 1.93) | 1.49 (1.31, 1.69) | 1.39 (1.20, 1.61) | 0.051 | 1.54 (1.27, 1.87) | <0.001 |
| Comparing broad-spectrum to narrow-spectrum antibiotics (events = 3063, person-years = 217,428) | | | | | | | |
| Any | 1.22 (1.10, 1.36) | 1.11 (1.00, 1.24) | 1.24 (1.10, 1.41) | 1.15 (1.01, 1.31) | 0.708 | 1.13 (1.04, 1.23) | 0.005 |
| Additional adjusting for infections[4] (events = 4577, person-years = 323,556) | | | | | | | |
| Any | 1.45 (1.33, 1.59) | 1.14 (1.03, 1.26) | 1.01 (0.91, 1.12) | 0.91 (0.81, 1.03) | 0.004 | 1.02 (0.82, 1.27) | 0.851 |
| 1-5 | 1.38 (1.26, 1.52) | 1.12 (1.01, 1.24) | 0.95 (0.86, 1.06) | 0.90 (0.80, 1.02) | 0.006 | 1.01 (0.81, 1.25) | 0.95 |
| 6-11 | 1.86 (1.60, 2.17) | 1.30 (1.10, 1.53) | 1.19 (1.01, 1.39) | 0.98 (0.81, 1.18) | 0.025 | 1.13 (0.86, 1.49) | 0.388 |
| 12+ | 2.43 (2.00, 2.95) | 1.62 (1.32, 1.99) | 1.70 (1.41, 2.05) | 1.28 (1.03, 1.59) | 0.122 | 1.44 (1.14, 1.81) | 0.002 |
| Adjusting for smoking and BMI with multiple imputation (events = 4577, person-years = 323,556) | | | | | | | |
| Any | 1.45 (1.33, 1.59) | 1.19 (1.08, 1.31) | 1.01 (0.91, 1.12) | 0.96 (0.86, 1.07) | 0.004 | 1.07 (0.87, 1.32) | 0.533 |
| 1-5 | 1.38 (1.26, 1.52) | 1.15 (1.04, 1.27) | 0.95 (0.86, 1.06) | 0.93 (0.83, 1.04) | 0.005 | 1.03 (0.84, 1.28) | 0.758 |
| 6-11 | 1.86 (1.60, 2.17) | 1.39 (1.19, 1.63) | 1.19 (1.01, 1.39) | 1.04 (0.88, 1.24) | 0.015 | 1.21 (0.91, 1.60) | 0.188 |
| 12+ | 2.43 (2.00, 2.95) | 1.82 (1.49, 2.22) | 1.70 (1.41, 2.05) | 1.44 (1.18, 1.76) | 0.105 | 1.62 (1.29, 2.04) | <0.001 |
| Early stage (events = 3475, person-years = 301,681) | | | | | | | |
| Any | 1.45 (1.30, 1.61) | 1.21 (1.09, 1.35) | 1.00 (0.89, 1.13) | 0.96 (0.84, 1.09) | 0.006 | 1.08 (0.85, 1.36) | 0.520 |
| 1-5 | 1.37 (1.22, 1.52) | 1.17 (1.04, 1.30) | 0.94 (0.84, 1.07) | 0.92 (0.80, 1.05) | 0.008 | 1.04 (0.82, 1.31) | 0.756 |
| 6-11 | 1.90 (1.61, 2.25) | 1.45 (1.22, 1.72) | 1.19 (0.99, 1.42) | 1.08 (0.89, 1.32) | 0.030 | 1.26 (0.94, 1.67) | 0.117 |
| 12+ | 2.43 (1.96, 3.01) | 1.85 (1.48, 2.30) | 1.62 (1.31, 2.00) | 1.44 (1.14, 1.81) | 0.122 | 1.63 (1.28, 2.09) | <0.001 |
| Two-year lag (events = 3835, person-years = 280,115) | | | | | | | |
| Any | 1.36 (1.23, 1.50) | 1.12 (1.01, 1.24) | 1.01 (0.90, 1.13) | 0.97 (0.86, 1.10) | 0.083 | 1.04 (0.91, 1.20) | 0.534 |
| 1-5 | 1.31 (1.18, 1.45) | 1.09 (0.98, 1.20) | 0.97 (0.86, 1.08) | 0.94 (0.83, 1.07) | 0.086 | 1.01 (0.88, 1.17) | 0.837 |
| 6-11 | 1.72 (1.45, 2.04) | 1.31 (1.10, 1.56) | 1.20 (1.01, 1.43) | 1.11 (0.92, 1.34) | 0.197 | 1.21 (1.03, 1.43) | 0.024 |
| 12+ | 1.97 (1.57, 2.48) | 1.51 (1.20, 1.91) | 1.46 (1.18, 1.82) | 1.29 (1.03, 1.63) | 0.344 | 1.40 (1.19, 1.65) | <0.001 |
| Restricted to antibiotics in the first 5 years (events = 1780, person-years = 166,259) | | | | | | | |
| Any | 1.40 (1.19, 1.65) | 1.20 (1.01, 1.42) | 1.14 (0.96, 1.36) | 1.01 (0.84, 1.22) | 0.193 | 1.11 (0.94, 1.30) | 0.225 |
| 1-5 | 1.32 (1.12, 1.57) | 1.16 (0.97, 1.38) | 1.08 (0.90, 1.29) | 0.97 (0.80, 1.18) | 0.186 | 1.07 (0.90, 1.27) | 0.456 |
| 6-11 | 1.54 (1.23, 1.92) | 1.25 (1.00, 1.58) | 1.30 (1.04, 1.62) | 1.13 (0.89, 1.44) | 0.547 | 1.19 (1.01, 1.41) | 0.035 |
| 12+ | 2.02 (1.49, 2.74) | 1.57 (1.15, 2.14) | 1.40 (1.04, 1.90) | 1.12 (0.81, 1.54) | 0.138 | 1.33 (0.95, 1.84) | 0.094 |
| Excluding antibiotics in the first 12 months after diagnosis (events = 3835, person-years = 280,115) | | | | | | | |
| Any | 1.51 (1.37, 1.67) | 1.29 (1.17, 1.43) | 1.32 (1.18, 1.48) | 1.25 (1.11, 1.42) | 0.720 | 1.28 (1.18, 1.38) | <0.001 |
| 1-5 | 1.43 (1.29, 1.58) | 1.24 (1.11, 1.37) | 1.26 (1.12, 1.41) | 1.21 (1.07, 1.37) | 0.822 | 1.23 (1.13, 1.33) | <0.001 |
| 6-11 | 2.09 (1.76, 2.48) | 1.62 (1.36, 1.93) | 1.57 (1.31, 1.88) | 1.44 (1.19, 1.74) | 0.379 | 1.53 (1.35, 1.75) | <0.001 |
| 12+ | 2.48 (1.99, 3.08) | 1.98 (1.59, 2.47) | 2.08 (1.68, 2.56) | 1.73 (1.38, 2.16) | 0.387 | 1.85 (1.58, 2.17) | <0.001 |

[a]Adjusted model contains age, year of diagnosis, deprivation, stage, grade, surgery, radiotherapy, chemotherapy, tamoxifen use (in year after diagnosis), aromatase inhibitor use (in year after diagnosis), Charlson comorbidities (before diagnosis), home replacement therapy (before diagnosis) and statin, aspirin, and metformin use (after diagnosis) and is based upon a complete case analysis, except where otherwise stated.
[b]P-value for heterogeneity comparing the adjusted hazard ratio for England and Wales. [3]Model additionally contains antibiotic use in the year before diagnosis. [4]Model additionally contains infection after diagnosis as a time-varying covariate.

recurred may have received additional treatment, including chemotherapy known to cause immunosuppression[29], which may have increased their risk of infection and use of antibiotics, but we did not have comprehensive data on treatments occurring later than 1 year after diagnosis. There was evidence of this confounding, as in sensitivity analyses with longer lags and restricting to antibiotics within the first 5 years of diagnosis, our associations were attenuated, which is consistent with this bias. Otherwise, breast cancer patients prescribed more antibiotics may be more generally frail, and previous studies have shown strong associations between frailty and infection[30] and, even in cancer-free populations, women frequently using antibiotics have been shown to have increased all-cause

**Table 6 | Subgroup analyses for the association between antibiotics and breast cancer-specific mortality**

| ABs | England | | Wales | | P for hetero[b] | Pooled adjusted[a] HR (95% CI) | P |
|---|---|---|---|---|---|---|---|
| | Unadjusted HR (95% CI) | Adjusted[a] HR (95% CI) | Unadjusted HR (95% CI) | Adjusted[a] HR (95% CI) | | | |
| Oestrogen receptor positive (events = 1784, person-years = 168,722) | | | | | | | |
| Any | 1.38 (1.18, 1.61) | 1.12 (0.95, 1.31) | 1.20 (1.03, 1.40) | 1.05 (0.88, 1.24) | 0.596 | 1.08 (0.96, 1.22) | 0.179 |
| 1-5 | 1.33 (1.13, 1.55) | 1.10 (0.93, 1.29) | 1.14 (0.97, 1.34) | 1.04 (0.87, 1.23) | 0.642 | 1.07 (0.95, 1.20) | 0.278 |
| 6-11 | 1.57 (1.20, 2.04) | 1.08 (0.82, 1.42) | 1.36 (1.08, 1.72) | 1.05 (0.82, 1.34) | 0.858 | 1.06 (0.88, 1.28) | 0.514 |
| 12+ | 2.37 (1.69, 3.32) | 1.77 (1.26, 2.49) | 1.81 (1.38, 2.38) | 1.26 (0.94, 1.69) | 0.135 | 1.48 (1.06, 2.06) | 0.023 |
| Oestrogen receptor negative (events = 719, person-years = 24,707) | | | | | | | |
| Any | 1.17 (0.94, 1.44) | 1.00 (0.80, 1.24) | 0.91 (0.71, 1.16) | 0.77 (0.59, 1.00) | 0.147 | 0.89 (0.69, 1.14) | 0.338 |
| 1-5 | 1.12 (0.90, 1.39) | 0.96 (0.77, 1.21) | 0.85 (0.66, 1.10) | 0.73 (0.56, 0.96) | 0.121 | 0.85 (0.65, 1.11) | 0.231 |
| 6-11 | 1.56 (1.01, 2.40) | 1.18 (0.75, 1.86) | 1.07 (0.66, 1.74) | 0.93 (0.56, 1.54) | 0.490 | 1.06 (0.76, 1.49) | 0.736 |
| 12+ | 2.62 (1.39, 4.92) | 1.94 (1.03, 3.66) | 2.68 (1.54, 4.67) | 1.95 (1.10, 3.46) | 0.988 | 1.95 (1.27, 2.98) | 0.002 |
| Tamoxifen or aromatase inhibitor treatment in the first year after diagnosis (events = 3163, person-years = 261,464) | | | | | | | |
| Any | 1.59 (1.42, 1.79) | 1.25 (1.11, 1.41) | 1.19 (1.05, 1.35) | 1.14 (0.99, 1.31) | 0.346 | 1.20 (1.10, 1.32) | <0.001 |
| 1-5 | 1.51 (1.34, 1.71) | 1.21 (1.07, 1.37) | 1.13 (0.99, 1.28) | 1.11 (0.96, 1.28) | 0.381 | 1.17 (1.06, 1.28) | 0.001 |
| 6-11 | 1.98 (1.66, 2.37) | 1.39 (1.16, 1.67) | 1.38 (1.15, 1.66) | 1.19 (0.97, 1.46) | 0.262 | 1.29 (1.11, 1.51) | <0.001 |
| 12+ | 2.52 (2.01, 3.15) | 1.75 (1.39, 2.20) | 1.90 (1.54, 2.36) | 1.59 (1.26, 2.01) | 0.571 | 1.67 (1.42, 1.97) | <0.001 |
| No tamoxifen or aromatase inhibitor treatment in the first year after diagnosis (events = 1414, person-years = 62,094) | | | | | | | |
| Any | 1.23 (1.06, 1.44) | 1.08 (0.92, 1.26) | 0.74 (0.62, 0.89) | 0.76 (0.63, 0.92) | 0.007 | 0.91 (0.65, 1.27) | 0.581 |
| 1-5 | 1.17 (1.01, 1.37) | 1.04 (0.88, 1.22) | 0.70 (0.59, 0.85) | 0.73 (0.60, 0.89) | 0.007 | 0.88 (0.62, 1.24) | 0.451 |
| 6-11 | 1.64 (1.22, 2.20) | 1.27 (0.94, 1.72) | 0.90 (0.64, 1.25) | 0.92 (0.65, 1.31) | 0.177 | 1.10 (0.80, 1.50) | 0.557 |
| 12+ | 2.49 (1.67, 3.71) | 1.93 (1.28, 2.90) | 1.44 (0.96, 2.16) | 1.20 (0.77, 1.87) | 0.124 | 1.53 (0.96, 2.44) | 0.071 |
| Under age 50 years at diagnosis (events = 997, person-years = 65,986) | | | | | | | |
| Any | 1.39 (1.14, 1.70) | 1.09 (0.89, 1.35) | 1.20 (0.95, 1.51) | 1.21 (0.94, 1.56) | 0.549 | 1.14 (0.97, 1.34) | 0.114 |
| 1–5 | 1.32 (1.07, 1.61) | 1.05 (0.85, 1.30) | 1.13 (0.89, 1.43) | 1.17 (0.91, 1.52) | 0.523 | 1.10 (0.93, 1.29) | 0.262 |
| 6–11 | 1.93 (1.40, 2.65) | 1.39 (1.00, 1.93) | 1.44 (1.03, 2.02) | 1.30 (0.90, 1.86) | 0.774 | 1.35 (1.06, 1.72) | 0.016 |
| 12+ | 2.18 (1.42, 3.34) | 1.46 (0.94, 2.27) | 2.13 (1.43, 3.18) | 1.85 (1.19, 2.87) | 0.452 | 1.64 (1.20, 2.24) | 0.002 |
| Over age 55 years at diagnosis (events = 3033, person-years = 206,626) | | | | | | | |
| Any | 1.44 (1.29, 1.62) | 1.18 (1.05, 1.32) | 0.94 (0.83, 1.06) | 0.87 (0.76, 1.00) | 0.001 | 1.01 (0.76, 1.36) | 0.924 |
| 1–5 | 1.38 (1.23, 1.54) | 1.14 (1.01, 1.29) | 0.89 (0.79, 1.01) | 0.85 (0.74, 0.98) | 0.002 | 0.99 (0.74, 1.32) | 0.929 |
| 6–11 | 1.78 (1.48, 2.14) | 1.28 (1.05, 1.55) | 1.05 (0.86, 1.28) | 0.90 (0.72, 1.12) | 0.017 | 1.08 (0.76, 1.52) | 0.683 |
| 12+ | 2.48 (1.96, 3.14) | 1.81 (1.42, 2.30) | 1.54 (1.22, 1.95) | 1.32 (1.03, 1.69) | 0.070 | 1.54 (1.13, 2.11) | 0.006 |

[a]Adjusted model contains age, year of diagnosis, deprivation, stage, grade, surgery, radiotherapy, chemotherapy, tamoxifen use (in year after diagnosis), aromatase inhibitor use (in year after diagnosis), Charlson comorbidities (before diagnosis), hormone replacement therapy (before diagnosis) and statin, aspirin, and metformin use (after diagnosis) and is based upon a complete case analysis, except where otherwise stated.
[b]P-value for heterogeneity comparing adjusted hazard ratio for England and Wales.

mortality[31]. In patients prescribed multiple antibiotics, we observed marked differences even in the risk of death from causes not attributable to breast cancer, which suggests the role of confounding by variables not related to breast cancer, and confounding by frailty may offer some explanation for these differences.

**Strengths and limitations**
Our study has many strengths. The study was much larger than previous studies and included 44,452 breast cancer patients with follow-up of up to 22 years. However, the average duration of follow-up may not have allowed us to capture later breast cancer-specific mortality. The study also included fewer breast cancer patients than in the pre-study sample size calculation, but as these calculations were conservative and the proportion using antibiotics was larger than anticipated, post-hoc power calculations would indicate we had sufficient power to detect the hazard ratios hypothesised. For example, with -70% using antibiotics and 4577 breast cancer-specific deaths, we would still have over 95% power to detect an HR of 1.2 in breast cancer-specific mortality in antibiotic users compared with non-users after diagnosis. A further strength was that our analyses were conducted in two independent data sources from England and Wales. This revealed some heterogeneity in the estimates, and in the main analysis, associations were consistently stronger in England compared with Wales, but associations were more similar for individual antibiotics and in some sensitivity analyses (e.g., when using longer lags or when excluding antibiotics in the first 6 months after diagnosis).

We ascertained antibiotics from prescribing records, eliminating recall bias. However, we would not have captured antibiotics prescribed in the hospital, which may have a broader spectrum, or antibiotics given in dental care. Further, our data sources do not capture information on adherence to antibiotics, but a previous UK study[32] showed that over 95% of patients prescribed an antibiotic took at least some antibiotic, and 75% finished the full course. Another study[33] showed that 70% of individuals prescribed an antibiotic for immediate use took at least some antibiotic, although some took a different antibiotic from that originally prescribed.

A limitation of our study was that, since it was observational, breast cancer patients were not randomly allocated to antibiotics after diagnosis, and consequently, recorded and unrecorded

**Table 7 | Sensitivity analyses for the association between antibiotics and different outcomes**

| ABs | England | | Wales | | P for hetero[b] | Pooled adjusted[a] HR (95% CI) | P |
|---|---|---|---|---|---|---|---|
| | Unadjusted HR (95% CI) | Adjusted[a] HR (95% CI) | Unadjusted HR (95% CI) | Adjusted[a] HR (95% CI) | | | |
| Main analysis: Breast cancer, underlying cause of death (events = 4577, person-years = 323,557) | | | | | | | |
| Any | 1.45 (1.33, 1.59) | 1.19 (1.08, 1.31) | 1.01 (0.91, 1.12) | 0.96 (0.86, 1.07) | 0.004 | 1.07 (0.87, 1.33) | 0.521 |
| 1-5 | 1.38 (1.26, 1.52) | 1.15 (1.05, 1.27) | 0.95 (0.86, 1.06) | 0.93 (0.83, 1.04) | 0.005 | 1.04 (0.84, 1.28) | 0.749 |
| 6-11 | 1.86 (1.60, 2.17) | 1.39 (1.19, 1.62) | 1.19 (1.01, 1.39) | 1.05 (0.88, 1.25) | 0.018 | 1.21 (0.92, 1.59) | 0.173 |
| 12+ | 2.43 (2.00, 2.95) | 1.81 (1.48, 2.20) | 1.70 (1.41, 2.05) | 1.45 (1.19, 1.78) | 0.127 | 1.62 (1.31, 2.01) | <0.001 |
| All-cause mortality (events = 9671, person-years = 323,557) | | | | | | | |
| Any | 1.47 (1.38, 1.58) | 1.26 (1.17, 1.35) | 1.01 (0.93, 1.08) | 0.98 (0.90, 1.06) | <0.001 | 1.11 (0.87, 1.42) | 0.410 |
| 1–5 | 1.36 (1.26, 1.45) | 1.19 (1.11, 1.28) | 0.92 (0.85, 0.99) | 0.92 (0.85, 1.00) | <0.001 | 1.05 (0.82, 1.34) | 0.711 |
| 6–11 | 2.03 (1.84, 2.24) | 1.53 (1.38, 1.70) | 1.19 (1.07, 1.32) | 1.09 (0.97, 1.22) | <0.001 | 1.29 (0.92, 1.81) | 0.136 |
| 12+ | 2.81 (2.50, 3.16) | 2.02 (1.79, 2.28) | 1.86 (1.66, 2.08) | 1.54 (1.36, 1.75) | 0.002 | 1.76 (1.36, 2.29) | <0.001 |
| Breast cancer anywhere on the death certificate (events = 5368, person-years = 323,556) | | | | | | | |
| Any | 1.47 (1.35, 1.60) | 1.23 (1.13, 1.34) | 1.00 (0.91, 1.10) | 0.97 (0.87, 1.08) | <0.001 | 1.09 (0.87, 1.38) | 0.449 |
| 1-5 | 1.39 (1.27, 1.52) | 1.18 (1.08, 1.29) | 0.94 (0.85, 1.04) | 0.93 (0.83, 1.04) | <0.001 | 1.05 (0.83, 1.32) | 0.692 |
| 6-11 | 1.96 (1.71, 2.25) | 1.47 (1.28, 1.69) | 1.20 (1.04, 1.40) | 1.09 (0.93, 1.28) | 0.006 | 1.27 (0.95, 1.70) | 0.106 |
| 12+ | 2.62 (2.20, 3.12) | 1.96 (1.64, 2.34) | 1.80 (1.52, 2.13) | 1.54 (1.28, 1.85) | 0.068 | 1.74 (1.38, 2.20) | <0.001 |
| Breast cancer is not the underlying cause of death (events = 5094, person-years = 323,556) | | | | | | | |
| Any | 1.50 (1.36, 1.65) | 1.28 (1.16, 1.42) | 1.00 (0.90, 1.12) | 0.93 (0.82, 1.06) | <0.001 | 1.09 (0.80, 1.49) | 0.569 |
| 1-5 | 1.33 (1.20, 1.47) | 1.19 (1.07, 1.32) | 0.88 (0.78, 0.98) | 0.86 (0.76, 0.98) | <0.001 | 1.01 (0.74, 1.38) | 0.927 |
| 6-11 | 2.15 (1.88, 2.46) | 1.58 (1.38, 1.82) | 1.17 (1.01, 1.35) | 1.02 (0.87, 1.20) | <0.001 | 1.27 (0.83, 1.95) | 0.274 |
| 12+ | 3.04 (2.61, 3.54) | 2.03 (1.73, 2.37) | 1.91 (1.64, 2.21) | 1.41 (1.20, 1.67) | 0.002 | 1.69 (1.19, 2.41) | 0.004 |
| Breast cancer not anywhere on death certificate (events = 4303, person-years = 323,556) | | | | | | | |
| Any | 1.48 (1.32, 1.65) | 1.25 (1.11, 1.39) | 1.01 (0.90, 1.14) | 0.93 (0.81, 1.06) | <0.001 | 1.08 (0.81, 1.44) | 0.619 |
| 1-5 | 1.30 (1.16, 1.46) | 1.15 (1.03, 1.30) | 0.88 (0.78, 1.00) | 0.86 (0.75, 0.99) | 0.002 | 1.00 (0.75, 1.33) | 0.994 |
| 6-11 | 2.08 (1.79, 2.41) | 1.51 (1.30, 1.76) | 1.16 (0.99, 1.35) | 0.99 (0.84, 1.18) | <0.001 | 1.23 (0.81, 1.86) | 0.332 |
| 12+ | 2.93 (2.48, 3.45) | 1.92 (1.62, 2.28) | 1.87 (1.59, 2.19) | 1.37 (1.15, 1.64) | 0.008 | 1.63 (1.17, 2.26) | 0.004 |

[a]Adjusted model contains age, year of diagnosis, deprivation, stage, grade, surgery, radiotherapy, chemotherapy, tamoxifen use (in year after diagnosis), aromatase inhibitor use (in year after diagnosis), Charlson comorbidities (before diagnosis), hormone replacement therapy (before diagnosis) and statin, aspirin, and metformin use (after diagnosis) and is based upon a complete case analysis, except where otherwise stated.
[b]P-value for heterogeneity comparing adjusted hazard ratio for England and Wales.

characteristics of antibiotic non-users and users, particularly those receiving 12 or more prescriptions, are likely to be different both at breast cancer diagnosis and after diagnosis when antibiotic prescriptions were received. We attempted to address these differences by adjusting for confounders captured within routinely collected electronic health records, but large differences persisted in mortality outcomes unrelated to breast cancer, suggesting residual confounding remains. Future studies of antibiotics and breast cancer outcomes should utilise prospective cohort designs to capture more detailed information on the outcome (including date of breast cancer recurrence and related treatments), the exposure (including reason for antibiotic use), and important covariates (such as frailty, the type and timing of infections after breast cancer diagnosis, and use of probiotics).

In this Registered Report, we observed similar breast cancer-specific mortality in breast cancer patients prescribed antibiotics after diagnosis, but higher breast cancer-specific mortality in breast cancer patients prescribed 12 or more antibiotics. These patients requiring multiple antibiotic courses are likely to have had multiple infections, have poorer general health, and are inherently likely to have a higher mortality risk. The attenuation of this association for multiple antibiotics after various adjustments, and a similar association with multiple antibiotics for deaths not attributable to breast cancer, highlights the role of confounding in the association between antibiotics and breast cancer-specific mortality, and it is not clear whether we were able to fully account for this in our analysis.

## Methods
### Data sources
The study used two independent data sources: Secure Anonymised Information Linkage (SAIL) Databank (Wales)[34], and QResearch (England)[35]. The SAIL Databank is a population-based data repository from Wales with linkages between datasets conducted on a unique identifier. The study used the following datasets from SAIL Databank: Welsh cancer registry data (the Welsh Cancer Intelligence Surveillance Unit), national mortality data (from the Annual District Death Extract), GP data (from Welsh Longitudinal General Practice Dataset), and hospital data (Patient Episode Dataset Wales). QResearch is a database of anonymised health records from England based on GP records (from the Optum computer system). The study used linkages (based upon encrypted NHS number) between the GP records and other sources, including mortality data (from Office of National Statistics), cancer registration data (from Public Health England cancer registration data), and hospital data (from Hospital Episode Statistics). These data sources were selected because they contain high-quality cancer registry data, mortality data, prescribing data, and detailed patient characteristics.

### Study population
Population-based cohorts of women newly diagnosed, from 2000 to 2017, with incident breast cancer (based upon ICD 10 code C50) of stage 1–3 were identified using cancer registry records in England and Wales (from the Welsh Cancer Intelligence and Surveillance Unit).

Patients previously diagnosed with other invasive cancer diagnoses (apart from non-melanoma skin cancer) were excluded. Patients were registered with a GP on the date of their breast cancer diagnosis and had a year of GP records prior to the date of their breast cancer diagnosis.

## Exposure

Oral antibiotic use was ascertained from electronic GP prescribing records in England and Wales based upon Section 5.1 of the British National Formulary (BNF)[36]. Oral antibiotics were categorised by drug class (based on BNF[36] classification) into the following types: penicillins, cephalosporins, tetracyclines, macrolides, clindamycin, sulphonamides, trimethoprim, metronidazole and tinidazole, quinolones, nitrofurantoin and methenamine, and other antibacterials. Antibiotics were also categorised by specificity (categorised as broad- or narrow-spectrum[37]).

## Outcome

The primary outcome was breast cancer-specific mortality (based upon ICD-10 code C50 breast cancer as the underlying cause of death) identified from 2000 up to 2021 from national mortality records. Sensitivity analyses investigated breast cancer-specific mortality based on breast cancer listed anywhere on the death certificate. Secondary analyses investigated all-cause mortality.

## Covariates

The following covariates were identified:

a) age and year of breast cancer diagnosis (from cancer registry records),
b) cancer stage and grade at diagnosis (from cancer registry records),
c) cancer treatment (surgery, radiotherapy, and chemotherapy from cancer registry and hospital records for the year after diagnosis),
d) hormone receptor status (England only, from cancer registry records),
e) hormone therapy (including tamoxifen and aromatase inhibitors, from GP prescribing records),
f) comorbidities (including the Charlson comorbidity conditions myocardial infarction, congestive heart failure, peripheral vascular disease, stroke, COPD, hemiplegia, dementia, liver disease, peptic ulcer disease, diabetes and chronic kidney disease, from GP records and hospital admissions),
g) medication use (medications potentially associated with breast cancer-specific mortality, such as aspirin[38], statins[39], metformin[40], and hormone replacement therapy[41] from GP prescribing records),
h) deprivation (using the 2011 Index of Multiple Deprivation in Wales and the 2011 Townsend deprivation score in England from GP records),
i) smoking status and BMI (from GP records), and
j) diagnosed infections (from GP records and hospital admissions).

## Statistical analysis

Table 1 provides a summary of all planned analyses. There is the possibility of information bias[42] caused by missing data. In general, where appropriate, we used multiple imputation to impute missing values[43]. Twenty imputed datasets were created, and as recommended, breast cancer-specific mortality status and the cumulative hazard[44] along with covariates were included in imputation models, and results were combined using Rubin's rules[43]. STATA version 18 (StataCorp, TX) was used for all analyses. Analyses were reported according to STROBE guidelines. At the time of writing the Stage 1 Registered Report, all of the data or evidence that was used to answer the research question existed but was inaccessible to the research team (consistent with a level 5 Registered Report). However, the research team had conducted

previous research in earlier breast cancer cohorts from SAIL Databank[45] and QResearch[35], but these data were obtained under licence for other purposes, and antibiotics were not investigated. The analyses commenced in December 2023 and were completed in September 2025.

## Main analysis

There were two main analyses of oral antibiotic use after breast cancer diagnosis: one part investigated the use of any antibiotic (any use and frequency of use), and one part investigated antibiotic use by type. In these main analyses, breast cancer patients with stage 1–3 disease were followed from 12 months after diagnosis to breast cancer-specific mortality (censoring on death from other causes, end of mortality follow-up, or end of GP records). Patients who died in the first 12 months after diagnosis were excluded, as it seems unlikely that antibiotic use after diagnosis could impact such deaths. Patients were required to have medication records for the year before cancer diagnosis. Patients using antibiotics in the year prior to diagnosis, who may have depleted microbiota at cancer diagnosis, were excluded. Antibiotic use after diagnosis was modelled as a time-varying covariate to avoid immortal time bias[46] and was lagged by 12 months[28] with antibiotic users compared with antibiotic non-users (described in Fig. 1). This 12-month lag will reduce reverse causation from antibiotics used for infections as part of end-of-life care. Sensitivity analyses, described below, were conducted by varying the duration of the lag (using a lag of 2 years and 3 years).

The first part of the main analysis investigated the use of any antibiotic (1 or more prescriptions compared with non-users) and frequent use of any antibiotic based upon the cumulative number of prescriptions (1–5, 6–11, 12 or more prescriptions compared with non-users). The second part of the main analysis investigated the use of antibiotics by type. Time-dependent Cox proportional hazards (PH) regression models were used to calculate Hazard Ratios (HRs), and 95% confidence intervals (CIs), for antibiotic use compared with non-use after breast cancer diagnosis adjusting for age at diagnosis (continuous), year of diagnosis (continuous), stage (in categories 1–3), grade (in categories 1–3), surgery (yes or no), radiotherapy (yes or no), chemotherapy (yes or no), hormone therapy use (any versus none, after diagnosis as time varying covariates), Charlson comorbidities (separately before diagnosis), hormone replacement therapy use (yes or no, before diagnosis), other medication use (including statin, aspirin and metformin, any versus none, after diagnosis as time varying covariates) and deprivation (in fifths). The PH assumptions were checked by visual inspection of log(−log) plots. A two-stage analysis procedure using random effects models was conducted to pool results across cohorts using the Hazard Ratios and corresponding standard errors in each cohort[47].

## Sensitivity analyses

A number of sensitivity analyses were conducted to explore the consistency of the association between antibiotic use and breast cancer-specific mortality. Despite the number of sensitivity analyses, we have not made any correction for multiple comparisons because none of these sensitivity analyses will be interpreted in isolation from each other. First, an analysis of antibiotic use after diagnosis was conducted, not excluding individuals using antibiotics in the year before diagnosis. Second, an analysis of antibiotic use after diagnosis was conducted, adjusting for antibiotic use in the 1 year before diagnosis. Third, an active comparator analysis was conducted comparing broad-spectrum with narrow-spectrum antibiotics as both broad and narrow-spectrum antibiotics may have similar indications, but narrow-spectrum antibiotics are likely to have less impact on the microbiome[48]. Fourth, an analysis was conducted of antibiotics prescribed after diagnosis and breast cancer-specific mortality, additionally adjusting for infections. Fifth, analyses were conducted to investigate potential reverse

causation: (a) an analysis was repeated restricted to early stage disease (stage 1–2); (b) separate analyses were conducted using a 2 year medication lag (starting follow-up at 24 months) and 3 year medication lag (starting follow-up at 36 months); (c) an analysis was conducted only investigating antibiotic use within the first 5 years after breast cancer diagnosis (starting follow-up at 5 years), and (d) an analysis was conducted excluding antibiotic use in the 6 months after diagnosis (starting follow-up at 18 months) and excluding antibiotic use in the year after diagnosis (starting follow-up at 24 months). Sixth, in England, analyses were repeated by hormone receptor status (as the immune system may play a greater role in the biology of oestrogen receptor negative disease[49]) and the analyses were repeated in both datasets by receipt of any endocrine therapy (tamoxifen or aromatase inhibitors) in the first 12 months after diagnosis, a proxy for oestrogen receptor status. Seventh, the main analysis was repeated, additionally adjusting for smoking and BMI, which are potential confounders as they may be associated with both antibiotic use[50] and cancer-specific mortality[51,52]. Multiple imputation and complete case analyses were conducted because we anticipated missing data for these variables, as in a previous study using UK GP records, BMI and smoking were missing for around 20% of breast cancer patients[35]. Eighth, as a proxy for menopausal status, the main analysis was repeated, restricted to women aged over 55 (a proxy for postmenopausal status) and under 50 (a proxy for premenopausal status). Finally, the analyses were repeated, investigating breast cancer-specific mortality based on breast cancer listed anywhere on the death certificate and all-cause mortality.

### Further details on preregistered analyses

In the sensitivity analysis of antibiotics and breast cancer-specific mortality, additionally adjusting for infections, infections were identified using previously published code lists from GP[53] and hospital records[54], and separate GP infection and hospital infection variables were included in models as time-varying covariates.

### Changes to preregistered analyses

Originally, based upon anticipated data availability, we had planned to include population-based cohorts of women diagnosed with breast cancer from 2000 to 2017 and followed for mortality up to 2021, but the final analysis included women diagnosed between 2000 to 2019 and followed up to July 2023 in both England and Wales. We had originally planned to only identify and analyse hormone receptor status in England, but using records from the Cancer Network Information System (CNIS) in Wales, we were also able to conduct subgroup analyses by hormone receptor status in Wales.

### Unplanned analyses (Stage 2 analyses)

In addition to the planned analyses of the Registered Report, we conducted some additional analyses to explore the specificity of the observed association between antibiotic use and breast cancer-specific mortality. Specifically, we repeated the main analysis using the outcomes of death where breast cancer was not the underlying cause and death where breast cancer was not mentioned on the death certificate.

### Sample size calculation

We estimated the Welsh breast cancer cohort would contain over 17,000 stage 1–3 breast cancer patients newly diagnosed from 2000 to 2017, in whom there would be 2125 cancer-specific deaths. In England, we estimated there would be over 60,000 stage 1–3 breast cancer patients in whom we would expect 7500 cancer-specific deaths, based upon Welsh cancer-specific mortality rates. Antibiotic prescribing prevalence was estimated from a previous case-control study[55]. In the year before breast cancer diagnosis, an estimated 30% of breast cancer patients received an antibiotic prescription; in a 5-year period, 55% of all breast cancer patients received an antibiotic prescription, and 4% received 12 or more antibiotic prescriptions.

Using Schoenfeld's method[56], based upon the numbers above, we would have over 95% power to detect a clinically important, relative 20% increase (i.e., a HR of 1.2) in breast cancer-specific mortality in antibiotic users compared with non-users after diagnosis. Similarly, in users of 12 or more antibiotics, we could detect a 25% increase in cancer-specific mortality, and by antibiotic class, we could detect a 20% increase in cancer-specific mortality with use of cephalosporins (the antibiotic type used in the mouse model experiments[17]) or penicillin (based upon 10% use of cephalosporins and 33% use of penicillin).

### Ethics

Ethical approval for the QResearch database is obtained annually from the East Midlands–Derby Research Ethics Committee (Ref. 18:/EM/0400). Approval for our analysis of the English data has been obtained from the QResearch Scientific Committee. Approval for analysis of the Welsh data has been obtained from the SAIL Databank Information Governance Review Panel.

### Reporting summary

Further information on research design is available in the Nature Portfolio Reporting Summary linked to this article.

## Data availability

The datasets from Wales (SAIL databank; Swansea University, https://saildatabank.com/) and England (QResearch; Queen Mary University of London, https://www.qresearch.org/) were obtained under strict data access conditions that allowed the study to be conducted but do not allow direct data sharing. However, the data analysed in this study would, in principle, be available to a researcher who applied to the data custodians and obtained the same approvals.

## Code availability

Relevant STATA code is available in the Supplementary Material.

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

## Acknowledgements

This work was supported by a project grant from Cancer Research UK (reference PRCPJT-Nov22\100017). The authors would also like to thank Mrs. Magaret Grayson, a Patient and Public Involvement (PPI) representative, for providing input into the research question, the study design, interpretation of the study, and lay summary materials. We acknowledge the contribution of EMIS practices that contribute to QResearch and Queen Mary University of London for expertise in developing and supporting the QResearch database. This project involves data derived from patient-level information collected by the NHS, as part of the care and support of cancer patients. The hospital, cancer, and mortality data are collated, maintained, and quality assured by the National Disease Registration Service, which is part of NHS England. Access to the data was facilitated by the NHS England Data Access Request service. NHS England bears no responsibility for the analysis or interpretation of the data. We would also like to acknowledge the support of SAIL Databank for facilitating access to the dataset from Wales.

## Author contributions

Acquired funding: C.R.C., Ú.Mc.M., S.A.M.cl., B.H., C.A.C.C., A.J.B., F.J.B. and J.H.C.; Study design: C.R.C., Ú.Mc.M., B.H., C.A.C.C., F.J.B. and J.H.C. Statistical analysis: C.R.C., C.A.C.C., F.J.B., S.B., A.J.H.L.S., D.T.C. and E.C.A. Clinical expertise: S.A.M.cl, A.J.B. and J.H.C. Writing first draft: C.R.C. Approval of final draft: C.R.C., Ú.Mc.M., S.A.Mc.I., B.H., C.A.C.C., A.J.B., F.J.B., J.H.C., S.B., A.J.H.L.S., D.T.C. and E.C.A.

## Competing interests

S.A.Mc.I. reports institutional funding for honoraria from Roche, MSD and AstraZeneca, for advisory boards from Roche, Lily and MSD, for talks from Roche and MSD, and research funding from Novartis. S.A.Mc.I. also reports personal funding for talks from B.D. Bard and conference travel and support from Roche and Lilly. A.J.B. reports institutional funding from Pfizer, MSD and Shionogi to support delivery of educational events. The remaining authors declare they have no relevant financial or non-financial interests.

## Additional information

[1]Centre for Public Health, Queen's University Belfast, Belfast, Northern Ireland, UK. [2]Wolfson Institute of Population Health, Queen Mary University of London, London, UK. [3]Nuffield Department of Primary Care Health Sciences, University of Oxford, Oxford, UK. [4]Division of Health Sciences, University of Warwick, Coventry, UK. [5]Breast Surgery Department, Belfast City Hospital, Belfast Health and Social Care Trust, Belfast, Northern Ireland, UK. [6]The Patrick G Johnston Centre for Cancer Research, Queen's University Belfast, Belfast, Northern Ireland, UK. [7]Belfast Health and Social Care Trust, Belfast City Hospital, Belfast, Northern Ireland, UK. [8]School of Pharmacy, Queen's University Belfast, Belfast, Northern Ireland, UK. ✉e-mail: c.cardwell@qub.ac.uk

