## [Transparent Peer Review file · Nature Communications]

Antibiotic use and survival from breast cancer: A population-based cohort study in England and Wales

Corresponding Author: Professor Christopher Cardwell

Editorial Note: This manuscript was submitted to Nature Communications as a Registered Report. The reviews and rebuttals for both the pre-study (Stage 1) and post-study (Stage 2) are included within this peer review file.

STAGE 1

Version 0:

Reviewer comments:

Reviewer #1

(Remarks to the Author)

I. Significance of the research question(s) for the field of study:

As stated by the authors, the use of antibiotics is high (and likely over-prescribed) and the prevalence of breast cancer is high. Because the survival rate for breast cancer is also high, it's important to understand factors that may be associated with survival, especially factors that may be modifiable such as antibiotic use. This is an important research question and the study approach has been well thought out and described.

II. Analysis and methodology:

The study design and data described are appropriate for the research question proposed. There are some limitations in the available data that should be considered:

1. An important limitation is the lack of receptor status data from one of the two data bases. If these data are indeed coming from a tumor registry, it is surprising these data are not available since they guide treatment decisions and are also associated with survival.
2. Is menopausal status available? How will menopausal status be estimated and considered in the analysis?
3. It would be helpful to know the completeness and validity of the data abstracted from "GP records", especially for the broader research community who may not be familiar with how these data are obtained. In particular, data such as alcohol use, smoking, and other social factors may not be adequately captured.
4. The authors state that smoking will be included, which is not consistently associated with risk of breast cancer. However, alcohol use and physical activity are associated with both risk and survival of breast cancer. Are these data available? If not, these are limitations to this analysis which should be acknowledged.
5. The title indicates that 2 cohorts will be used – suggesting that this is an important element to the study. However, other than sample size (which appears adequate), why is this important? One cohort is missing data on receptor status, a very important data element. Further, how likely is it that the results from Wales versus England will differ?
6. Table 1 lists "Main Analyses" – however, this is far more than "main analyses". Analyses #4, for example, provides a list of dozens of models. While I appreciate the comprehensive list, I believe that this many models could result in a multiple comparisons problem. The authors should either decide on the most important question to address, or provide a way to account for multiple comparisons

III. Biases:

1. An important bias has been overlooked by the authors and should be considered: information bias will likely be a problem in the analysis and interpretation of these data. Women who are “healthier” will have less data than women who are less healthy – thus creating a differential bias. One example is BMI, which is associated both with risk of breast cancer and breast cancer survival, as well as other important conditions such as hypertension and diabetes. Those with high BMI may have more encounters with the health care system, and thus more data, than those with lower BMI.

I am not a proponent of data imputation, as proposed by the authors. Can you demonstrate that imputed data can help address this bias (I think it's unlikely given that behavioral factors that are unmeasured contribute to missingness)?

2. Screening. Are screening data available in these data sets? It is important to account for use of mammography and other breast screening modalities because breast cancer recurrence and second breast cancer rates will differ among women who are not screening regularly compared to those who are.

Reviewer #2

(Remarks to the Author)

The aim of the research to provide ‘precise estimates of the association between antibiotic use after diagnosis, by frequency and antibiotic class, and survival from breast cancer’ is of interest and significance. There are very few published studies to date that have examined this association and with increasing burden of antibiotics globally and most countries demonstrating an increasing trend in the incidence of breast cancer this is an important area of study.

The logic and rationale of the study is provided in the introduction with reference to previous studies, including animal studies.

Is there any more evidence to support the hypotheses under study? Only one study (reference 13) is referred suggesting less diverse gut microbiota in breast cancer.

Some of the evidence supporting the hypothesis is not strong – what are the limitations if any of the studies included in the introduction?

A study examining the association of antimicrobials in women with triple negative breast cancers (Ransohoff et al 2021) also refers to antibiotics altering immune cells and lymphocytes in the blood of women with breast cancer on antibiotics. It might be useful to refer to alternative mechanisms in the rationale for the study if appropriate.

Overall the methodology and analysis plan are generally sound and feasible. The proposed study will use two large databases, one in England and one in Wales.

However, there are some additional suggestions to improve clarity and provide more detail to sufficiently replicate the study procedures.

Did the authors consider all sources of potential for bias e.g. detection bias – will those on antibiotics be more likely to attend their GP or sicker individuals, how might this introduce bias and what can be done to account for this?

Will there be temporal changes in use and type of antibiotics used in Wales/England over the window of study e.g. 2000-2017? This may be quite variable over time with many advances/changes in the types of antibiotics used. This is not really mentioned.

It would be helpful to have more details provided on the SAIL and Qresearch data sources and linkages to cancer registration included in the methods. How representative of the population, what proportion of the population is covered by these data sources, etc.

Covariates – how were these selected for inclusion? Were these based on evidence, and/or what was available in the database? If from evidence, references to support should be included. There are others referred to in Ransohoff et al 2021, which are not mentioned here e.g. race, ethnicity. Other medication use refers to including statin, aspirin and metformin – are there other medications, why these specific medicines of interest?

Some details on the covariates would be welcome. For example, for comorbidities (including Charlson comorbidity) – what was included and diagnosed infections (from GP records and hospital admissions) – what does this include? Types of surgery etc. Also, how will these be handled in the analysis – for example, is a count of infections or a category y/n for diagnosed infection to be used (overall or individual types)? Is there confounding in how sick people are more likely to be prescribed antibiotics?

Reference group used in the case of each covariate (in case of categorical covariates) should be provided and the type of variables (category, continuous, etc.) in each case.

The exposure categorisation uses number of prescriptions cumulatively, which appears quite a crude method. Did the authors consider alternative approaches that include the type/dose/duration of exposure? What about use in ophthalmology, intranasal, etc. were these also included? A full list of the antibiotics included could be provided in an appendix.

In the exposure – antibiotic exposure is considered but not the control group (no prescriptions at any time post-diagnosis)? If the Proportional Hazard assumption fails what will be the next steps for analysis?

It might be worth including some information on the power/sample size for new user design (as a sensitivity analysis) but probably not required for all sensitivity analysis.

There are several additional analyses proposed (11+ sensitivity analyses) – what is the rationale behind each of these? These will be repeated analyses on the same datasets. Is it possible to limit to fewer key sensitivity analyses?

Sample size calculation - why are the rates not available for England. Is the assumption of similar rates in Wales and England reasonable?

The study will have power to detect a relative 10% increase (i.e. a HR of 1.1) in breast cancer-specific mortality between antibiotic users compared with non-users. This appears a fairly modest association – is this of clinical relevance?

Table 1 provides a summary of main analysis. However, there is no specific reasons for each sensitivity analysis. Perhaps

this table might be moved to an appendix with the reasons behind analyses included.

Do the authors anticipate any ceiling effects (at the upper end) in number of prescriptions categorised as: none, 1 to 5, 6 to 11, 12 or more prescriptions?

Refer to Cox proportional hazards model in statistical methods as the abbreviation PH appears in reference to assessing 'PH assumptions'.

A two-stage analysis will be used to pool results from both databases - this needs some further explanation. Has this been previously applied to survival data?

What reporting guidelines that will be referred to in the final publication e.g. STROBE, etc. ?

Acknowledgements and funding sources are not provided.

Reviewer #3

(Remarks to the Author)

This manuscript describes the protocol evaluation possible effects of antibiotics in patients with breast cancer. There are the following comments:

-Title refers to independent cohorts. It is unclear what independent means. While Sail and Qresearch are regionally distinct, patients could be included in both if they moved between regions.

-The study inclusion is "with stage 1 to 3 incident breast cancer (based upon ICD 10 code C50)". It should be clarified that staging information would come from the cancer registry (and not from ICD10 as currently implied). As we observed in a similar study (although with different aims: https://papers.ssrn.com/sol3/papers.cfm?abstract_id=3382398), the staging information in cancer registries is often incomplete (although I do not recall the specific numbers for breast cancer). Good to consider an alternative approach if staging information is missing more than rarely.

-The protocol states "chemotherapy from cancer registry and hospital records". Cancer registry data may be incomplete with respect to chemotherapy data, and hospital records do not have prescription data; they may be sometimes recorded under procedures. Good to consider the possible effects of incomplete data.

-The protocol states "Patients who died in the first 18 months after diagnosis will be excluded as it seems unlikely that antibiotic use after diagnosis could impact upon such deaths". This is OK although it would be very useful to explore whether patients with history with frequent antibiotic use would have similar mortality in months after diagnosis compared to women without (assessment of confounding).

-Table 1 lists plan for various analyses, which is OK. One comment is around the interpretation column stating e.g. "HR for any antibiotic use > 1.15 in both analysis provides support for detrimental impact of antibiotic use". Not sure an arbitrary cut-off (which also varies between analyses) would change the interpretation, plus it does not consider statistical significance (e.g. a non-significant HR of 1.15 would not support a detrimental effect). Rather than an arbitrary cut-off, it would be more consider the Bradford Hill criteria (e.g. dose response). While the protocol does consider immortal time bias, there is little consideration and plans for bias analyses. The main concern around adverse effects of repeated antibiotic use is confounding (i.e., patients with this use are different). Good to include bias analyses (e.g. looking at HRs shortly after cancer diagnosis).

-the protocol states "an analysis of antibiotics use after diagnosis will be conducted adjusting for antibiotic use in the in the 2 years before diagnosis". Not sure how one can or should adjust the exposure for prior exposure, especially if it could be relevant for the possible causal mechanism (on microbiota). Plus, frequent antibiotic exposure is often not an one-off but correlated over time (<https://pubmed.ncbi.nlm.nih.gov/32114981/>)

-the exposure definition is unclear, particularly around antibiotic exposure prior to diagnosis (if considered, patients with short follow-up prior to diagnosis should be excluded). Will everyone start with non-exposure in the time-dependent analysis or will antibiotic exposure prior to the diagnosis be considered? Reading the methods, outcome follow-up starts at months 18 and with all antibiotics in first 6 months excluded (as well as any prior antibiotic exposure); this indicates that everyone starts with non-exposure at months 18, which will then change based on antibiotics given in months 6+. This exposure definition seems fundamentally wrong if the possible causal mechanism is considered; patients with a history of antibiotics prior to month 6 may have a depleted microbiota and can not be considered non-exposed. It is important to reconsider this incorrect exposure definition.

Version 1:

Reviewer comments:

Reviewer #1

(Remarks to the Author)

The authors did not address the majority of the limitations that I pointed out in my review. In particular, I asked for more

description of the "GP records" and how complete these data are, but I don't see that added to the revised manuscript.

I maintain that information bias is an important limitation of this study, but the authors did not address this in the revised paper, and in fact, minimized it in their response letter.

There is literature to support the association between BMI and frequency of antibiotic prescription, however, my comment on this was largely dismissed by the authors in their reply.

The lack of data on physical activity and alcohol use is also an important limitation as both are associated with use of antibiotics, overall survival, and risk of breast cancer. I did not see where this was added to the revised paper.

Reviewer #2

(Remarks to the Author)

[Reviewer #2 had no further comments]

Reviewer #3

(Remarks to the Author)

The authors have made several changes to the manuscript which is appreciated. However, there is still a major struggle and inconsistency with their exposure definitions. I have several comments:

-They state "...However, guidelines on studies of the effects of exposures after cancer diagnosis on survival (JNCI 105:1456–1462) additionally recommend (1) restricting the analysis to individuals unexposed to the medication before cancer diagnosis and (2) adjusting for precancer medication use. For completeness, we have included both these analyses as sensitivity analyses in the Registered Report...".

=> If this guideline is to be followed, this exclusion should not merely be a sensitivity analysis. Plus, exclusion of prior frequent antibiotic users from the analysis will likely reduce any effect given the correlation over time of frequent antibiotic use.

=> Plus, if I interpret this guideline correctly, it would mean that for an analysis of the effects on cancer treatments of smoking, one would need to restrict the analysis to people starting smoking after start of cancer therapy. Or adjust for prior smoking history which would mean adjusting away any effect. This seems bizarre.

-They state "...we have ignored antibiotics before diagnosis because these are clearly not modifiable and we have ignored antibiotics in the first 6 months after diagnosis because we think these will reflect initial treatment and will probably not be modifiable..".

=> in a retrospective study, nothing is modifiable, so this argument is not convincing. In clinical practice, we can of course do something about frequent antibiotic in patients presenting today etc.

=> as stated previously, the underlying biological mechanism may be effects on the microbiota. It makes little sense to this reviewer that everyone in this study (given the lagged exposure definition of 12 months) will start in the non-exposed category. Those with use in the first 6 months or use before the cancer diagnosis may not have unaffected microbiota and cannot be considered unexposed. It may of course make sense to exclude outcomes in the first 6 months (as more likely to be related to the cancer severity rather than effects of treatments). But the exposure definition should reflect actual exposure.

=> the proposed study design would make little sense, in the opinion of this reviewer, if the exposure of interest would be smoking exposure. In such an analysis, one would clearly consider prior smoking history and not assign everyone as non-smoker in the first 12 months of follow-up, irrespective of their prior history.

=> this is not about adding a few sensitivity analyses but rather about having a robust exposure definition which is consistent with the underlying biological mechanism. The current exposure definition is in my view is incorrect and does not overcome the confounding challenge.

-They state "...Also, we will conduct an analysis of antibiotic use before diagnosis which removes some of the potential biases of the analyses of antibiotics after diagnoses and allows inclusion of all deaths after cancer diagnosis=..". I am not sure why an exposure definition that includes all prior antibiotic exposure at some period before would need to include all deaths. It is sensible to exclude deaths within 6 months of diagnosis (as more related to cancer severity). But with follow-up starting at months 6, one could have e.g. a time-dependent exposure classification (e.g. antibiotic use in XX years before as determined at each time point of follow-up).

Reviewer #4

(Remarks to the Author)

I am commenting only on whether the paper adequately meets the requirements of a Registered Report format. The authors do a very good job of making clear the rationale for the study, the proposed hypotheses and associated tests. In particular, the authors state their criteria for interpretation for each test in Table 1. Some further recommendations:

1) The authors improved the clarity of Table 1 by indicating what is the main analyses, what are sensitivity analyses, and what are exploratory analyses. However, I would suggest the authors not list a p-value threshold for the exploratory analyses. These by definition are hypothesis generating, not testing, so p-values do not have much value here. Instead the authors can focus on reporting the effect size and uncertainty estimates. This would also be true for any other exploratory analyses the authors might choose to add to the final report when they explore the dataset.

2) The authors have multiple main and sensitivity tests, yet do not do anything to control for multiple comparisons. I defer to the other reviewers on the statistical soundness of this and recommend to the authors to include in the manuscript their rationale for not including anything. Right now, this information is in the rebuttal letter and it would be highly beneficial from a transparency perspective to include it in the manuscript so readers know what the authors decisions were before analysis.

Version 2:

Reviewer comments:

Reviewer #3

(Remarks to the Author)

This is to outline my response to the Authors' comments.

-The authors have misrepresented the article that apparently formed their basis for their exposure definition. This article is "Threats to Validity of Nonrandomized Studies of Postdiagnosis Exposures on Cancer Recurrence and Survival Jessica Chubak, Denise M. Boudreau, Heidi S. Wirtz, Barbara McKnight, Noel S. Weiss. In contrast to what is stated, this is not a guideline but work by a research group (although this article is very sensible, it can not be described as a guideline and should not be presented as such).

-The responses by the authors with respect to my feedback on the exposure definition are generally quite poor scientifically as they are not well argued. For example, it is stated that "We think this exposure definition better captures the underlying mechanism". Science is not about beliefs but about providing reasonable rationale for e.g. design choices. Do the authors believe that prior history of antibiotics is irrelevant for the health of microbiota??? Of course, one can disagree on design choices but a rationale for a choice should be provided.

-It would be useful for the authors to consider the use of DAGs to better delineate possible causal pathways.

-It is reasonable to consider lagging of exposure due the possibility of undetected cancer recurrence causing infections and antibiotic use (protopathic bias). However, I do not think it is appropriate to consider this lagging time as 'non-exposed' time simply because one believes that the exposure during this 'lagging' time has no causal impact. For the same reason, one may want to consider antibiotic exposure during the first 12 months as possibly this exposure could have causal effects (there is no reason to believe that only exposure to antibiotics after 12 months+lagging will have effects on microbiota and that before not. Rather that allocating these more complex periods of time to 'non-exposure', it is far more reasonable to consider additional exposure periods. This would allow the non-exposure time to be non-exposed rather than being a bit truly non-exposed and a bit which relies on an assumption of no causal effect.

-A related comment relates to the handling of patients with prior antibiotic exposure (which is not infrequent). The latest approach is "Patients using antibiotics in the year prior to diagnosis will be allocated to a past antibiotic use category and remain in this category for the duration of follow-up.". As this past exposure will be a mix of true past exposure (if no further antibiotics were used) and current exposure (with new antibiotic being used), this past exposure group will not be informative, and the study provide no evidence for this group of patients. Again, in my view a better more detailed exposure classification is very much preferred (which would reflect the complexity of the antibiotic exposures and uncertain causal pathways).

Reviewer #4

(Remarks to the Author)

I've been asked to comment specifically on the design table and whether the sensitivity analysis should be seen as exploratory analysis and thus removed entirely from the table. The authors addressed all my previous comments sufficiently.

For the design table. The authors should edit the table to remove the mention of exploratory analysis since it only includes main and sensitivity analyses. As far as whether to include the sensitivity analyses, I'd recommend keeping them included since they can be specified in advanced, which they are, including how to interpret the results. Opposed to unplanned analyses, which can not, by definition, but are good to state that there is interest in exploring these relationships during stage 1 of the Registered Report.

STAGE 2

Version 3:

Reviewer comments:

Reviewer #2

(Remarks to the Author)

The study report will be of significance to the field as there are few studies that have considered the association of antibiotic use in women with breast cancer on breast-cancer specific mortality.

The authors refer to Unplanned analyses (Stage 2 analyses) which involve sensitivity analysis around the definition of outcome, and are reported separately in the results section.

The conclusions appear appropriate - the authors do not over interpret the significance of the findings as they suggest residual confounding might still exist.

There are some minor comments for clarification and that might improve the final manuscript.

Line 31-32: The aim in abstract refers to 'we investigated whether breast cancer patients using oral antibiotics had increased breast cancer-specific mortality', however, the results also report on all-cause mortality (which is not the main analysis but described as sensitivity analysis in the methods).

Page 8 – line 187 please insert 'pooled' before 'adjusted hazard ratio'

Line 218 'patients prescribed 12 or more antibiotics was reduced' – perhaps refer to the revised HR or % risk.

Line 230-231 when referring to other studies, the authors state 'no significant association with any antibiotic use after diagnosis (HR=1.39 95% CI 0.93, 2.29),' was shown even though the HR is quite large – this is likely due to the small study size, perhaps the authors refer to the strengths or limitations of these studies by way of critique.

The paper mentioned by one reviewer Domzaridou et al was not mentioned in the discussion?

Discussion limitations – what about the fact that over the counter medications are not captured? Also, there were no implications of the findings mentioned in the discussion – is it possible to discuss the consequences or application of the findings to practice or policy?

(Remarks on code availability)

The Stata code provide was not in the form of a README file, and limited comments but with the appropriate 'cohort' file available and variables listed it should be possible to run (I did not attempt to run the code on the data). The models provided in the code were (i) a model with only antibiotic use and (ii) a model with antibiotic use (presumably y/n) and a list of covariates. The description of the 'antibyn' variable and other covariates used were not described in detail in the accompanying table.

Reviewer #3

(Remarks to the Author)

I thank the authors for this work.

My comments on Journal's questions:

-Whether the data are able to test the authors' proposed hypotheses by passing the approved outcome-neutral criteria (such as absence of floor and ceiling effects or success of positive controls)

=> yes

- Whether the Introduction, rationale and stated hypotheses are the same as the approved Stage 1 submission (I have appended a copy of the in-principle accepted Stage 1 manuscript and Supplementary Information)

=> yes

- Whether the authors adhered to the registered experimental procedures

=> Important to state when analyses were conducted: were all analyses conducted after registration of the protocol or were some analyses conducted before?

- Whether any unregistered exploratory analyses added by the authors are justified, methodologically sound, and informative

=> OK

- Whether the authors' conclusions are justified given the data

=>The conclusion states that "the attenuation of the associations in sensitivity analyses, and similar findings for other causes of death, suggest this increase may be attributable to residual confounding". I am not sure about the strength of this conclusion. Firstly, the sensitivity analyses around the effects of other causes of death ignore the real challenges of using cause-of-death data to attribute exact of cause of death. Death certificates tend to overrepresent immediate complications. This analysis of death due to other causes lacked a robustness of e.g. negative control design. Secondly, several sensitivity analyses seem to support the primary hypothesis (e.g. change of lag time), so conclusion is too categorical. Thirdly, it is well known that the more subgroup analyses one conducts (e.g. found in randomised trials), the muddier the results become. Fourthly, Thirdly, there is the fundamental question of what to conclude if primary and relevant sensitivity analyses vary. The conclusion, I think, is that no firm conclusions are possible.

Additional comment:

=>Good to conduct literature search on previous work that has explored the same research question. For example, I could find no reference and discussion of related work: DOI: 10.3390/currenol30090614

(Remarks on code availability)

No expert in stata code

Reviewer #4

(Remarks to the Author)

The authors do a very nice job of describing their results from their stage 1 Registered Report, including any deviations and unplanned analyses. My only comment is in the introduction - the authors might want to end of introduction it might be good to change the tense (e.g., "The study will investigate..." to 'The study does investigate....')

(Remarks on code availability)

Editorial Note: STAGE 1

Reviewers' Comments:

Reviewer #1 (Remarks to the Author):

I. Significance of the research question(s) for the field of study:

As stated by the authors, the use of antibiotics is high (and likely over-prescribed) and the prevalence of breast cancer is high. Because the survival rate for breast cancer is also high, it's important to understand factors that may be associated with survival, especially factors that may be modifiable such as antibiotic use. This is an important research question and the study approach has been well thought out and described.

*We thank the reviewer for their comments.

II. Analysis and methodology:

The study design and data described are appropriate for the research question proposed. There are some limitations in the available data that should be considered:

1. An important limitation is the lack of receptor status data from one of the two data bases. If these data are indeed coming from a tumor registry, it is surprising these data are not available since they guide treatment decisions and are also associated with survival.

*We agree with the reviewer that the lack of estrogen receptor status in one of the datasets is a weakness and will mention this in the weakness section of the final manuscript. We don't think that estrogen receptor status is likely to be a major confounder because it is not clear why it would be associated with antibiotic use. We will have estrogen receptor status available in England, the larger of our two datasets. Further, to address this weakness we have added to the Registered Report an additional sensitivity analysis stratifying by use of endocrine therapy (tamoxifen and aromatase inhibitors) which is likely to reflect estrogen receptor status and can be conducted in both datasets.

2. Is menopausal status available? How will menopausal status be estimated and considered in the analysis?

*We agree that lack of menopausal status is a weakness as it is not routinely recorded in either dataset. We have added a sensitivity analysis stratified by age to the Registered Report to address this weakness. Based upon cut offs suggested in previous studies (Maturitas; 67: 60–66), we propose conducting an analysis in over 55s to investigate a subgroup in which the majority of women will be postmenopausal and an analysis in under 50s to investigate a subgroup in which the majority of women will premenopausal.

3. It would be helpful to know the completeness and validity of the data abstracted from "GP records", especially for the broader research community who may not be familiar with how these data are obtained. In particular, data such as alcohol use, smoking, and other social factors may not be adequately captured.

*In the main analysis the majority of the included GP data will be complete including comorbidities and medication use. In the sensitivity analysis adjusting for additional confounders we anticipate some missing data for smoking of around 10 to 15% and for BMI of around 20% based upon our previous use of SAIL GP data. A study comparing GP smoking data with survey data showed smoking status was relatively well recorded in primary care but with some misclassification of ex-smokers as non-smokers if quitting occurred at an early age or a long time ago (BMJ Open 2014; 4:e004958.).

We do not intend to adjust for alcohol use as we have concerns about the completeness and validity of the GP alcohol data and we do not think alcohol is likely to be a large confounder.

4. The authors state that smoking will be included, which is not consistently associated with risk of breast cancer. However, alcohol use and physical activity are associated with both risk and survival of breast cancer. Are these data available? If not, these are limitations to this analysis which should be acknowledged.

*We agree this is a weakness that we cannot adjust for these variables because neither are well recorded in GP records, and in the final manuscript we will discuss this limitation. It is unclear how much confounding influence these will have as we do not know the extent to which they will be associated with antibiotic use and we will adjust for other lifestyle variables (such as deprivation, BMI and smoking) which is likely to attenuate any confounding influence of these variables. It is also reassuring that in a recent meta-analysis alcohol intake was not associated with breast cancer prognosis (Int J Cancer 2023;152:616–634) and although physical activity was associated with survival, authors have suggested this could reflect reverse causation (Breast 2019;44:144-152).

5. The title indicates that 2 cohorts will be used – suggesting that this is an important element to the study. However, other than sample size (which appears adequate), why is this important? One cohort is missing data on receptor status, a very important data element. Further, how likely is it that the results from Wales versus England will differ?

*We think this independent replication is an important element of the study. First, As the author acknowledges, it does increase power, but more importantly if we observe an association in one dataset which replicates in the other this would reduce the risk of Type 1 error. Whilst we agree that there are similarities between the two datasets, many features of how the data are coded, recorded and linked are different and some treatment practices may vary across England and Wales. Consequently, we think that replicating the analysis in the two datasets would confer a robustness to the findings.

6. Table 1 lists “Main Analyses” – however, this is far more than “main analyses”. Analyses #4, for example, provides a list of dozens of models. While I appreciate the comprehensive list, I believe that this many models could result in a multiple comparisons problem. The authors should either decide on the most important question to address, or provide a way to account for multiple comparisons

*We have now highlighted clearly in Table 1 and the text of the Registered Report which analyses are the main analysis and which are sensitivity analyses (in which we are testing how consistent the associations are after varying the assumptions of the main analysis), or are exploratory analyses (and for hypothesis generating purposes). We are keen to retain the analyses but we will not interpret any of the sensitivity analyses in isolation, so we don’t think there is a need for corrections for multiple comparisons.

III. Biases:

1. An important bias has been overlooked by the authors and should be considered: information bias will likely be a problem in the analysis and interpretation of these data. Women who are “healthier” will have less data than women who are less healthy – thus creating a differential bias. One example is BMI, which is associated both with risk of breast cancer and breast cancer survival, as well as other important conditions such as hypertension and diabetes. Those with high BMI may have more encounters with the health care system, and thus more data, than those with lower BMI. I am not a proponent of data imputation, as proposed by the authors. Can you demonstrate that imputed data

can help address this bias (I think it's unlikely given that behavioral factors that are unmeasured contribute to missingness)?

*We think that this bias is not likely to have a large impact on the main exposure (antibiotics), outcome (breast cancer-specific mortality) or main confounders (cancer stage, grade, other diagnoses, or medications use). However, we agree that information bias may be a problem for lifestyle confounders for instance BMI. BMI is not in our main model partly because we do not think it will have a strong association with antibiotics and partly because we are aware of the difficulty using routinely recorded BMI in GP records. The reviewer is correct that the imputation model will make assumptions about the missing mechanism and these assumptions are largely untestable. We will explore the characteristics of those with missing BMI compared with available BMI. However, we agree that it is possible that BMI may not be Missing At Random conditional on the variables to which we have access. In light of the reviewer's concerns, we have added a complete case analysis to the Registered Report for BMI and smoking but we will discuss the potential for bias in this sensitivity analysis in the final manuscript.

2. Screening. Are screening data available in these data sets? It is important to account for use of mammography and other breast screening modalities because breast cancer recurrence and second breast cancer rates will differ among women who are not screening regularly compared to those who are.

*Unfortunately screening data are not available in our dataset. However, our primary outcome is breast cancer-specific mortality which will not be as influenced by screening as for instance recurrence. Further, all women will usually have a mammogram every year, for at least 5 years, after treatment for breast cancer and only at that point will they be released back into the population-based screening programme, so at least initially we would anticipate that the surveillance would be similar within the included women.

Reviewer #2 (Remarks to the Author):

The aim of the research to provide 'precise estimates of the association between antibiotic use after diagnosis, by frequency and antibiotic class, and survival from breast cancer' is of interest and significance. There are very few published studies to date that have examined this association and with increasing burden of antibiotics globally and most countries demonstrating an increasing trend in the incidence of breast cancer this is an important area of study.

*We thank the reviewer for their comments.

The logic and rationale of the study is provided in the introduction with reference to previous studies, including animal studies.

Is there any more evidence to support the hypotheses under study? Only one study (reference 13) is referred suggesting less diverse gut microbiota in breast cancer.

*We have added further evidence to this line of the Registered Report highlighting a systematic review which included a number of studies investigating the microbiome and breast cancer (Front Oncol. 2023; 13: 1144021).

Some of the evidence supporting the hypothesis is not strong – what are the limitations if any of the studies included in the introduction?

*We think the main weakness of the animal experiments is whether these will translate to humans so we have added this weakness to the Registered Report and included a reference on how animal experiments sometimes do not translate to humans (JACC Basic Transl Sci 2019;4:845-854). A weakness of the previous epidemiology studies is sample size: the conducted studies were relatively small making it difficult to estimate associations precisely.

A study examining the association of antimicrobials in women with triple negative breast cancers (Ransohoff et al 2021) also refers to antibiotics altering immune cells and lymphocytes in the blood of women with breast cancer on antibiotics. It might be useful to refer to alternative mechanisms in the rationale for the study if appropriate.

*We think if we observe an association between antibiotics and breast cancer prognosis we will certainly add these potential explanations to the final manuscript.

Overall the methodology and analysis plan are generally sound and feasible. The proposed study will use two large databases, one in England and one in Wales.

However, there are some additional suggestions to improve clarity and provide more detail to sufficiently replicate the study procedures.

Did the authors consider all sources of potential for bias e.g. detection bias – will those on antibiotics be more likely to attend their GP or sicker individuals, how might this introduce bias and what can be done to account for this?

*We will not directly account for GP attendance but we think various aspects of the analysis should account for sicker patients receiving antibiotics. First, we adjust for a range of comorbidities and medication use. Second, antibiotic use will be investigated with a 12 month lag thus ignoring antibiotics in the 12 months before death and reducing any reverse causation caused by sickness or frequent exposure to health care professionals related to end of life care. Also, the primary outcome is breast cancer-specific mortality which will have a weaker association with general sickness than all-cause mortality.

Will there be temporal changes in use and type of antibiotics used in Wales/England over the window of study e.g. 2000-2017? This may be quite variable over time with many advances/changes in the types of antibiotics used. This is not really mentioned.

*We do not think there have been large changes in patterns of antibiotic use between 2000 to 2017 but there will have been some changes. We are including year of diagnosis in the models which should account for this and prevent any bias caused by differences in the year of diagnosis of antibiotic users compared with non-antibiotic users. We could, at the discretion of the editor, add analyses stratified by year of diagnosis to the Registered Report.

It would be helpful to have more details provided on the SAIL and Qresearch data sources and linkages to cancer registration included in the methods. How representative of the population, what proportion of the population is covered by these data sources, etc.

*We agree this would be useful so we have added the following paragraph to the Registered Report: "The SAIL Databank is a population-based data repository from Wales with linkages between datasets conducted on a unique identifier. The study will use the following datasets from SAIL Databank: the Welsh cancer registry (the Welsh Cancer Intelligence Surveillance Unit), national mortality data (from the Annual District Death Extract), GP data (from Welsh Longitudinal General

Practice Dataset) and hospital data (Patient Episode Dataset Wales). QResearch is a database of anonymised health records from England based upon GP records (from the EMIS computer system). The study will use linkages (based upon encrypted NHS number) between the GP records and other sources including mortality data (from Office of National Statistics), cancer registration data (from Public Health England cancer registration data) and hospital data (from Hospital Episode Statistics).”

Covariates – how were these selected for inclusion? Were these based on evidence, and/or what was available in the database? If from evidence, references to support should be included. There are others referred to in Ransohoff et al 2021, which are not mentioned here e.g. race, ethnicity. Other medication use refers to including statin, aspirin and metformin – are there other medications, why these specific medicines of interest ?

*The main model contains the following variables: age at diagnosis, year of diagnosis, stage, grade, surgery, radiotherapy, chemotherapy, hormone therapy use (after diagnosis as time varying covariates), Charlson comorbidities (separately before diagnosis), hormone replacement therapy use (before diagnosis), other medication use (including statin, aspirin and metformin, after diagnosis as time varying covariates) and deprivation. Many of these variables were chosen based upon clinical expertise because they are known to be associated with cancer prognosis including age, year, stage, grade, surgery, radiotherapy and chemotherapy. We included Charlson comorbidities (specifically myocardial infarction, congestive heart failure, peripheral vascular disease, stroke, COPD, hemiplegia, dementia, liver disease, peptic ulcer disease, diabetes and chronic kidney disease) as a measure of general health. We included deprivation of home address as a measure of lifestyle/social factors which could be related to antibiotic use and cancer prognosis. The specific medications listed above were included because previous studies have shown associations between these variables and breast cancer prognosis including statin (Int J Cancer 2016;139:1281-8), metformin (Oncologist 2015;20:1236-44), aspirin (Oncologist, doi: 10.1093/oncolo/oyad186), and HRT before diagnosis (Breast Cancer 2017;24:643-657). We have now added references to these studies in the Registered Report to provide evidence for why we selected these medications. Unfortunately, we have some concerns about the completeness and accuracy of the ethnicity data routinely collected in the health records and hence we do not plan to adjust for ethnicity, however we don't think this is likely to be a major confounder.

Some details on the covariates would be welcome. For example, for comorbidities (including Charlson comorbidity) – what was included and diagnosed infections (from GP records and hospital admissions) – what does this include? Types of surgery etc. Also, how will these be handled in the analysis – for example, is a count of infections or a category y/n for diagnosed infection to be used (overall or individual types)? Is there confounding in how sick people are more likely to be prescribed antibiotics?

*In the main model we will use the following Charlson comorbidities: myocardial infarction, congestive heart failure, peripheral vascular disease, stroke, COPD, hemiplegia, dementia, liver disease, peptic ulcer disease, diabetes and chronic kidney disease. We have added these conditions to the Registered Report. We will adjust for cancer registry recorded surgery, radiotherapy or chemotherapy. If detailed information is available by type of surgery, we may conduct an additional analysis adjusting for type of cancer treatment.

In the sensitivity analysis in which we adjust for infection, we will investigate all hospital diagnosed infections and GP diagnosed infections using published code lists (e.g. BMJ Open. 2019 Apr 3;9(4):e026251 and Pharmacoepidemiol Drug Saf. 2018 Oct;27(10):1147-1150). The codes used will be made available in the final manuscript. We will conduct an analysis adjusting for infection. In this analysis of antibiotics after diagnosis, we will create time varying covariates for these infections and

indications using a lag of 1 year and we will conduct analysis additionally adjusting for these infections along with other confounders.

As described above, the main outcome is cancer-specific mortality which is less sensitive to general sickness, we will use lags on our outcome definition to reduce reverse causation from sickness near death and adjust for general ill health (using comorbidities from the Charlson index) in the model. Various sensitivity analyses will also control for sickness such as analyses adjusting for infection and the active comparator analysis (which will be conducted comparing broad-spectrum with narrow-spectrum antibiotics, thus both the broad and narrow-spectrum antibiotics will be prescribed antibiotics for infection, but the narrow-spectrum antibiotic users are likely to have less impact on the microbiome).

Reference group used in the case of each covariate (in case of categorical covariates) should be provided and the type of variables (category, continuous, etc.) in each case.

*In the Registered Report, I have highlighted the categories for the included variables.

The exposure categorisation uses number of prescriptions cumulatively, which appears quite a crude method. Did the authors consider alternative approaches that include the type/dose/duration of exposure? What about use in ophthalmology, intranasal, etc. were these also included? A full list of the antibiotics included could be provided in an appendix.

*We think that number of prescriptions is sufficiently accurate to capture long term usage, and that consistently calculating days of use based upon dose, pack size and direction may be difficult because this information may not be complete and completeness may differ between datasets, particularly for pack size and direction. We agree that it is a weakness that some antibiotics will be missed (e.g. prescribing in hospital ophthalmology clinics) but the vast majority of antibiotic prescribing in England ~80% is via primary care (see Public Health England. English Surveillance Programme for Antimicrobial Utilisation and Resistance (ESPAUR). October 2018.). We will include all oral antibiotics listed which are taken from Section 5.1 of the British National Formulary (BNF). This is now highlighted in the Registered Report and we will provide a full list of included antibiotics used in the final manuscript.

In the exposure – antibiotic exposure is considered but not the control group (no prescriptions at any time post-diagnosis)?

*Yes, antibiotic users are compared with antibiotic non-users using a time varying co-variate with a 12 month lag. We have clarified this in the methods of the Registered Report.

If the Proportional Hazard assumption fails what will be the next steps for analysis?

*Should the PH assumption fail for a covariate we would run a stratified Cox regression model (BJC 89:605–611) stratifying on this covariate (this would necessitate categorising a continuous covariate). Should the PH assumption fail for antibiotics we would use accelerated failure time models which do not require the PH assumption (BJC 89:431–436). We now discuss this in the Registered Report where we state: The Cox PH assumptions will be checked by visual inspection of log(-log) plots and should the PH assumption fail for a confounder a stratified cox model will be applied, or should the PH assumption fail for antibiotics an accelerated failure time approach will be adopted.

It might be worth including some information on the power/sample size for new user design (as a sensitivity analysis) but probably not required for all sensitivity analysis.

*We agree this is an important analysis and have added to the Registered Report a sample size calculation based upon the new user design. The following line has been added : “Also, in the new antibiotic user analysis (restricting the cohort to antibiotic non-users in the 2 years before diagnosis, conservatively assuming 50% did not use antibiotics in the 2 years before diagnosis) we would have 95% power to detect a relative 15% increase (i.e. a HR of 1.15) in breast cancer-specific mortality in antibiotic users after diagnosis compared with non-users.”

There are several additional analyses proposed (11+ sensitivity analyses) – what is the rationale behind each of these? These will be repeated analyses on the same datasets. Is it possible to limit to fewer key sensitivity analyses?

*We are keen to retain these analyses as each tests assumptions of our main analysis and we think will add robustness to the main findings. We have now clearly labelled these as Sensitivity or Exploratory Analyses in the Registered Report.

Sample size calculation - why are the rates not available for England. Is the assumption of similar rates in Wales and England reasonable ?

*The sample size is based upon the number of breast cancer-specific deaths in the breast cancer cohorts. Based upon an older Welsh breast cancer cohort we were able to calculate the numbers of breast cancer specific deaths in Wales. Cancer registries often publish figures for relative survival which is similar but not the same as breast cancer-specific, so we thought it was better to extrapolate the Wales rates onto the English cohort size. We think it is probably reasonable that rates of breast cancer-specific deaths in England and Wales would be similar.

The study will have power to detect a relative 10% increase (i.e. a HR of 1.1) in breast cancer-specific mortality between antibiotic users compared with non-users. This appears a fairly modest association – is this of clinical relevance?

*We agree this is a relatively small increase but breast cancer is common, antibiotics are widely used and death from breast cancer is clearly important to reduce as much as possible. Importantly, this sample size allows us to obtain a precise estimate of the association and allows us power to investigate the dose response and different antibiotic classes, particularly those implicated by the animal experiments.

Table 1 provides a summary of main analysis. However, there is no specific reasons for each sensitivity analysis. Perhaps this table might be moved to an appendix with the reasons behind analyses included.

*Table 1 is a required element of the Registered Report. We now highlight in this table which analyses are main analysis, which are sensitivity analyses and which are exploratory analyses.

Do the authors anticipate any ceiling effects (at the upper end) in number of prescriptions categorised as: none, 1 to 5, 6 to 11, 12 or more prescriptions?

*We think, if the association is real, we would anticipate a dose response across the categories.

Refer to Cox proportional hazards model in statistical methods as the abbreviation PH appears in reference to assessing ‘PH assumptions’.

*In the text of the Registered Report we now refer to this as the “Cox PH model” as suggested.

A two-stage analysis will be used to pool results from both databases - this needs some further explanation. Has this been previously applied to survival data?

*Yes we have clarified this in the Registered Report, basically we are extracting the hazard ratios and corresponding standard errors in each cohort and using a basic random effects model (as used in meta-analysis) to pool results across the two datasets (see Stat Med 2001;20:2115-30)

What reporting guidelines that will be referred to in the final publication e.g. STROBE, etc. ?

*We agree STROBE would be most appropriate and we now state “Analysis will be reported according to STROBE guidelines.”

Acknowledgements and funding sources are not provided.

*These will be added to the final manuscript. The study is funded by Cancer Research UK.

Reviewer #3 (Remarks to the Author):

This manuscript describes the protocol evaluation possible effects of antibiotics in patients with breast cancer. There are the following comments:

-Title refers to independent cohorts. It is unclear what independent means. While SAIL and QResearch are regionally distinct, patients could be included in both if they moved between regions.

*We have now clarified in the Registered Report that patients must have at least one year of GP records prior to the date of their breast cancer diagnosis. So, a patient registered at GP in England and diagnosed with breast cancer who then moved to Wales would be censored on the date their English GP records ended and could not be eligible to enter the Wales cohort as they were not at a Welsh GP when they were diagnosed. The Registered Report now states: “Patients must be registered at a GP on the date of their breast cancer diagnosis and have a year of GP records prior to the date of their breast cancer diagnosis.”

-The study inclusion is “with stage 1 to 3 incident breast cancer (based upon ICD 10 code C50)”. It should be clarified that staging information would come from the cancer registry (and not from ICD10 as currently implied). As we observed in a similar study (although with different aims: https://papers.ssrn.com/sol3/papers.cfm?abstract_id=3382398), the staging information in cancer registries is often incomplete (although I do not recall the specific numbers for breast cancer). Good to consider an alternative approach if staging information is missing more than rarely.

*We thank the reviewer for the reference which does provide some evidence of increased breast cancer-specific mortality in antibiotic users before diagnosis, but did not investigate breast cancer-specific mortality after diagnosis. Once published we will add this to the Discussion in the final manuscript.

We agree the current sentence is misleading and have rearranged the sentence in the Registered Report to clarify that staging will come from the cancer registry. We are confident that we will have enough breast cancer patients with available stage data as we anticipate that stage will be 85% complete in Wales based upon cancer registry reported figures and our experience with an earlier cohort, and QResearch preliminary counts were provided by stage.

-The protocol states “chemotherapy from cancer registry and hospital records”. Cancer registry data may be incomplete with respect to chemotherapy data, and hospital records do not have prescription data; they may be sometimes recorded under procedures. Good to consider the possible effects of incomplete data.

*We agree and we will discuss thoroughly the implications of potentially missing data in the final manuscript.

-The protocol states “Patients who died in the first 18 months after diagnosis will be excluded as it seems unlikely that antibiotic use after diagnosis could impact upon such deaths”. This is OK although it would be very useful to explore whether patients with history with frequent antibiotic use would have similar mortality in months after diagnosis compared to women without (assessment of confounding).

*We may have misunderstood but the analysis does include an analysis of antibiotic use in the year before diagnosis which will investigate all deaths after breast cancer diagnosis (i.e. this analysis will not exclude deaths in the 18 months after breast cancer diagnosis). We have added additional analyses (to the Registered Report) in which follow-up will start at 6, 12 and 24 months.

-Table 1 lists plan for various analyses, which is OK. One comment is around the interpretation column stating e.g. “HR for any antibiotic use > 1.15 in both analysis provides support for detrimental impact of antibiotic use”. Not sure an arbitrary cut-off (which also varies between analyses) would change the interpretation, plus it does not consider statistical significance (e.g. a non-significant HR of 1.15 would not support a detrimental effect). Rather than an arbitrary cut-off, it would be more consider the Bradford Hill criteria (e.g. dose response). While the protocol does consider immortal time bias, there is little consideration and plans for bias analyses. The main concern around adverse effects of repeated antibiotic use is confounding (i.e., patients with this use are different). Good to include bias analyses (e.g. looking at HRs shortly after cancer diagnosis).

*Table 1 and the included cut-offs are a required element of a Registered Report at Nature Communications. We agree that the interpretation will be more nuanced. We agree that the statistical significance will be important and evidence of a dose response will be very important. So we have added statistical significance to Table 1 in the Registered Report for the main analyses. For instance, the main analysis of any antibiotic now reads: “HR for any antibiotic use > 1.1 (and $P < 0.05$) provides support for increased cancer-specific mortality in antibiotic users” and we have the caveat “but further analysis of exposure-response and sensitivity analyses necessary.”

The cut-offs change between analyses reflecting the power for the different analyses. We have slightly less power for specific antibiotic classes and for 12 or more antibiotics because the proportion of individuals using specific antibiotic classes or 12 or more antibiotics will be lower.

We agree that investigating HRs by time from diagnosis could be informative. In the analysis of antibiotics after diagnosis we already include analysis that start at 12, 18 and 24 months. In the analysis of antibiotics before diagnosis, in which follow-up starts at diagnosis, we have added additional analyses (to the Registered Report) in which follow-up will start at 6, 12 and 24 months.

We think along with these a number of our analyses will control for bias for instance analysis which adjust for infection, stratify by infection and the active comparator analysis. In this analysis, we will compare users of broad-spectrum with users of narrow-spectrum antibiotics, as both groups are receiving antibiotics they are likely to be more similar but the narrow-spectrum antibiotic users are likely to have less impact on the microbiome.

-the protocol states “an analysis of antibiotics use after diagnosis will be conducted adjusting for antibiotic use in the in the 2 years before diagnosis”. Not sure how one can or should adjust the exposure for prior exposure, especially if it could be relevant for the possible causal mechanism (on microbiota). Plus, frequent antibiotic exposure is often not an one-off but correlated over time (<https://pubmed.ncbi.nlm.nih.gov/32114981/>)

*In the main analysis we do not adjust for antibiotics prior to diagnosis as we agree with the reviewer there is an argument not to adjust for antibiotic use before diagnosis. However, guidelines on studies of the effects of exposures after cancer diagnosis on survival (JNCI 105:1456–1462) additionally recommend (1) restricting the analysis to individuals unexposed to the medication before cancer diagnosis and (2) adjusting for precancer medication use. For completeness, we have included both these analyses as sensitivity analyses in the Registered Report.

-the exposure definition is unclear, particularly around antibiotic exposure prior to diagnosis (if considered, patients with short follow-up prior to diagnosis should be excluded). Will everyone start with non-exposure in the time-dependent analysis or will antibiotic exposure prior to the diagnosis be considered? Reading the methods, outcome follow-up starts at months 18 and with all antibiotics in first 6 months excluded (as well as any prior antibiotic exposure); this indicates that everyone starts with non-exposure at months 18, which will then change based on antibiotics given in months 6+. This exposure definition seems fundamentally wrong if the possible causal mechanism is considered; patients with a history of antibiotics prior to month 6 may have a depleted microbiota and can not be considered non-exposed. It is important to reconsider this incorrect exposure definition.

*In light of the confusion about the exposure definition we have added a diagram (Figure 1) to the Registered Report to explain the analysis of antibiotic exposure after diagnosis.

The reviewer is largely correct. Antibiotics in the 6 months after diagnosis are ignored. Everyone starts as unexposed and once they use an antibiotic they become exposed (after the 12 month lag) and consequently follow-up starts at 18 months. We understand the reviewers concerns about ignoring antibiotics before diagnosis and in the first 6 months after diagnosis. The reason we have ignored these antibiotics is because the goal of our main analysis is to investigate antibiotics which are potentially modifiable – we have ignored antibiotics before diagnosis because these are clearly not modifiable and we have ignored antibiotics in the first 6 months after diagnosis because we think these will reflect initial treatment and will probably not be modifiable. Obviously, we are making assumptions and we will explore the impact of these exclusion for instance in one sensitivity analysis we will not exclude any antibiotics after diagnosis and in another sensitivity analysis we will restrict the analysis to individuals who have not had an antibiotic in the 2 years before diagnosis (and hence will not have depleted microbiota). Also, we will conduct an analysis of antibiotic use before diagnosis which removes some of the potential biases of the analyses of antibiotics after diagnoses and allows inclusion of all deaths after cancer diagnosis, but arguably this information may be less clinically relevant. At the discretion of the editor we would rethink the exclusion of antibiotics in the 6 months after diagnosis in the main analysis.

Reviewers' Comments:

Reviewer #1 (Remarks to the Author):

The authors did not address the majority of the limitations that I pointed out in my review. In particular, I asked for more description of the "GP records" and how complete these data are, but I don't see that added to the revised manuscript.

*We think in a Stage 1 report it is hard to discuss missingness as we don't yet know the level of missingness of the covariates in our cohorts. The two variables of most concern are smoking and BMI. A previous QResearch case-control study included data on smoking and BMI. So, I have added the following lines to the Registered Report to highlight the degree of missingness we anticipate for these variables: "Multiple imputation and complete case analyses will be conducted because we anticipate missing data for these variables as in a previous study using UK GP records BMI and smoking were missing for approximately 20% of breast cancer patients"

We think it is more straightforward to discuss the degree of missingness and the implications of this in the final manuscript once we know exactly how complete variables are.

I maintain that information bias is an important limitation of this study, but the authors did not address this in the revised paper, and in fact, minimized it in their response letter.

*We apologise to the reviewer it was not our intention to minimize his concern and we agree with the reviewer that information bias is a concern. We now specifically mention information bias in the Stage 1 report and have added a reference for information bias. In a Stage 1 report there is not an obvious section in which to discuss potential limitations. Certainly, in the final manuscript we will discuss in detail the potential for information bias – at which, also, the degree of missingness in the data will be known.

There is literature to support the association between BMI and frequency of antibiotic prescription, however, my comment on this was largely dismissed by the authors in their reply.

*We have added a reference to the Stage 1 report highlighting the association between BMI and antibiotic use and BMI and cancer specific mortality. We previously added an additional sensitivity analysis (using complete case to adjust for

BMI and smoking) to attempt to address these variables. We will also more fully discuss these biases in the final manuscript.

The lack of data on physical activity and alcohol use is also an important limitation as both are associated with use of antibiotics, overall survival, and risk of breast cancer. I did not see where this was added to the revised paper.

*We agree with the reviewer these variables would be a valuable addition but in a Stage 1 report it is not clear where to discuss variables to which we will not have access. However, we will discuss these variables in more detail in the final manuscript.

Reviewer #2 (had no further comments)

Reviewer #3 (Remarks to the Author):

The authors have made several changes to the manuscript which is appreciated. However, there is still a major struggle and inconsistency with their exposure definitions. I have several comments:

-They state "...However, guidelines on studies of the effects of exposures after cancer diagnosis on survival (JNCI 105:1456–1462) additionally recommend (1) restricting the analysis to individuals unexposed to the medication before cancer diagnosis and (2) adjusting for precancer medication use. For completeness, we have included both these analyses as sensitivity analyses in the Registered Report...".

=> If this guideline is to be followed, this exclusion should not merely be a sensitivity analysis. Plus, exclusion of prior frequent antibiotic users from the analysis will likely reduce any effect given the correlation over time of frequent antibiotic use.

=> Plus, if I interpret this guideline correctly, it would mean that for an analysis of the effects on cancer treatments of smoking, one would need to restrict the analysis to people starting smoking after start of cancer therapy. Or adjust for prior smoking history which would mean adjusting away any effect. This seems bizarre.

*The methods suggested in the guidelines are intended for studies of the impact of exposures after cancer diagnosis (such as medication use) on outcomes, to attempt to inform cancer survivors (JNCI 105:1456–1462). The authors do acknowledge that these methods have strengths and weaknesses and we share some of the reviewer's concerns about these approaches. We have proposed a change to the exposure definition below in line with the reviewer's recommendations. However, we would be keen to retain the analyses recommended in the guidelines as a sensitivity analysis, as these approaches may provide reassurance to some readers who are aware of these guidelines.

-They state "...we have ignored antibiotics before diagnosis because these are clearly not modifiable and we have ignored antibiotics in the first 6 months after diagnosis because we think these will reflect initial treatment and will probably not be modifiable..".

=> in a retrospective study, nothing is modifiable, so this argument is not convincing.

In clinical practice, we can of course do something about frequent antibiotic in patients presenting today etc.

=> as stated previously, the underlying biological mechanism may be effects on the microbiota. It makes little sense to this reviewer that everyone in this study (given the lagged exposure definition of 12 months) will start in the non-exposed category.

Those with use in the first 6 months or use before the cancer diagnosis may not have unaffected microbiota and cannot be considered unexposed. It may of course make sense to exclude outcomes in the first 6 months (as more likely to be related to the cancer severity rather than effects of treatments). But the exposure definition should reflect actual exposure.

=> the proposed study design would make little sense, in the opinion of this reviewer, if the exposure of interest would be smoking exposure. In such an analysis, one would clearly consider prior smoking history and not assign everyone as non-smoker in the first 12 months of follow-up, irrespective of their prior history.

=> this is not about adding a few sensitivity analyses but rather about having a robust exposure definition which is consistent with the underlying biological mechanism. The current exposure definition is in my view is incorrect and does not overcome the confounding challenge.

*We thank the reviewer for the detailed description of his concerns. In light of these concerns, we propose altering the main analysis in line with his suggestion. We will restrict the cohort to individuals who have records in the year before diagnosis. Individuals who have used antibiotics in the year before diagnosis will enter the cohort as antibiotic users prior to diagnosis – and remain in this category. Follow-up of the cohort will start at 12 months after cancer diagnosis. As previously, individuals who use antibiotics after diagnosis will be considered users after a 12 month lag. We think this exposure definition better captures the underlying mechanism. We have updated the protocol to utilise this definition and have amended the sample size calculation, so it is now based upon this exposure definition. We have retained the original analysis as a sensitivity analysis.

-They state “..Also, we will conduct an analysis of antibiotic use before diagnosis which removes some of the potential biases of the analyses of antibiotics after diagnoses and allows inclusion of all deaths after cancer diagnosis=..”. I am not sure why an exposure definition that includes all prior antibiotic exposure at some period before would need to include all deaths. It is sensible to exclude deaths within 6 months of diagnosis (as more related to cancer severity). But with follow-up starting at months 6, one could have e.g. a time-dependent exposure classification (e.g. antibiotic use in XX years before as determined at each time point of follow-up).

*This analysis has been removed from the Stage 1 report as it was exploratory. However, we will conduct this analysis and add it to the final manuscript. We are keen to include all deaths in the analysis of antibiotic use before diagnosis. We think this is a strength of this analysis in that it avoids the need to exclude deaths after diagnosis over an arbitrary period. It is possible that the critical period for exposure to antibiotics is earlier in tumour development and if this leads to an increase in deaths shortly after diagnosis, analyses including all deaths we think would be desirable. We will include a sensitivity analysis of the medication use before diagnosis in which follow-up will start at 6 months.

Reviewer #4 (Remarks to the Author):

I am commenting only on whether the paper adequately meets the requirements of a Registered Report format. The authors do a very good job of making clear the rationale for the study, the proposed hypotheses and associated tests. In particular, the authors state their criteria for interpretation for each test in Table 1. Some further recommendations:

1) The authors improved the clarity of Table 1 by indicating what is the main analyses, what are sensitivity analyses, and what are exploratory analyses. However, I would suggest the authors not list a p-value threshold for the exploratory analyses. These by definition are hypothesis generating, not testing, so p-values do not have much value here. Instead the authors can focus on reporting the effect size and uncertainty estimates. This would also be true for any other exploratory analyses the authors might choose to add to the final report when they explore the dataset.

*In line with recommendations from the editor we have removed all exploratory analyses from the Stage 1 report.

2) The authors have multiple main and sensitivity tests, yet do not do anything to control for multiple comparisons. I defer to the other reviewers on the statistical soundness of this and recommend to the authors to include in the manuscript their rationale for not including anything. Right now, this information is in the rebuttal letter and it would be highly beneficial from a transparency perspective to include it in the manuscript so readers know what the authors decisions were before analysis.

*We have removed a considerable number of exploratory analyses from the Stage 1 report. We now explicitly state in the Stage 1 report our rationale for not correcting for multiple comparisons. We state “Despite the number of sensitivity analyses we will not make any correction for multiple comparisons because none of these sensitivity analyses will be interpreted in isolation.”

Reviewers' Comments:

Reviewer #3 (Remarks to the Author):

This is to outline my response to the Authors' comments.

-The authors have misrepresented the article that apparently formed their basis for their exposure definition. This article is "Threats to Validity of Nonrandomized Studies of Postdiagnosis Exposures on Cancer Recurrence and Survival Jessica Chubak, Denise M. Boudreau, Heidi S. Wirtz, Barbara McKnight, Noel S. Weiss. In contrast to what is stated, this is not a guideline but work by a research group (although this article is very sensible, it can not be described as a guideline and should not be presented as such).

*We agree with the reviewer. In the "Response to reviewers" document we referred to this article as a guideline but actually it is guidance/recommendations and not an actual guideline. We have not referred to this article as a guideline in the Stage 1 Report and hence have not made any changes to the manuscript.

-The responses by the authors with respect to my feedback on the exposure definition are generally quite poor scientifically as they are not well argued. For example, it is stated that "We think this exposure definition better captures the underlying mechanism". Science is not about beliefs but about providing reasonable rationale for e.g. design choices. Do the authors believe that prior history of antibiotics is irrelevant for the health of microbiota??? Of course, one can disagree on design choices but a rationale for a choice should be provided.

*We think it is possible that prior history of antibiotics could lead to depleted microbiota before diagnosis and hence could obscure the true association between antibiotics after diagnosis and cancer-specific mortality. Hence, in line with the reviewer's comments in the previous round of reviews we had allocated individuals using antibiotics in the year prior to diagnosis to a separate category of past antibiotic use however given his comments in this round of reviews, and as discussed below, we now propose simply excluding these individuals from the main analysis. We have now added a line explaining the rationale for this which reads: "Patients using antibiotics in the year prior to diagnosis, who may have depleted microbiota at cancer diagnosis, will be excluded".

-It would be useful for the authors to consider the use of DAGs to better delineate possible causal pathways.

*We agree, when interpreting the results, we will use DAGs to consider causal pathways.

-It is reasonable to consider lagging of exposure due the possibility of undetected cancer recurrence causing infections and antibiotic use (protopathic bias). However, I do not think it is appropriate to consider this lagging time as 'non-exposed' time simply because one believes that the exposure during this 'lagging' time has no causal impact. For the same reason, one may want to consider antibiotic exposure during the first 12 months as possibly this exposure could have causal effects (there is no reason to believe that only exposure to antibiotics after 12 months+lagging will have effects on microbiota and that before not. Rather than allocating these more complex periods of time to 'non-exposure', it is far more reasonable to consider additional exposure periods. This would allow the non-exposure time to be non-exposed rather than being a bit truly non-exposed and a bit which relies on an assumption of no causal effect.

*In our analysis we think a lag of 1 year is reasonable because it would seem unlikely biologically for antibiotics to precipitate a recurrence and breast cancer-specific mortality more quickly than this. We agree though there is debate around what duration of lag is appropriate and in sensitivity analysis we investigate longer lags.

The reviewer suggests categorising antibiotic use as: non-use, initial antibiotic use (between the date of first use and 12 months after first use), and antibiotic use (12 months after the date of use). We

are concerned about how this approach would handle antibiotic use immediately before death. For instance, if a month before death from cancer a cancer patient received their first antibiotic this death would be allocated to the initial antibiotic use category. It seems unlikely that this death was related to their antibiotic use so assigning this death to the initial antibiotic use category would lead to an artificially high risk of death in the initial antibiotic use category and an artificially low risk of death in the non-antibiotic use category. In contrast, in the current analysis we propose patients start as non-users and then become antibiotic users 12 months after the date of use, consequently this death would be correctly allocated to the antibiotic non-users.

The reviewer also raises concerns about excluding antibiotic use in the first 6/12 months after breast cancer diagnosis. However, this concern was raised and addressed in the previous round of reviews. In the previous round of reviews the reviewer was concerned that antibiotic use before diagnosis and in the 6 months after diagnosis could deplete microbiota and hence obscure associations between antibiotic use after diagnosis and cancer-specific mortality. We have amended the study in two ways to address this. First, as described above, we dropped users of antibiotics in the year prior to diagnosis and second we no longer exclude prescriptions in the 6 months after diagnosis. We think the current analysis addresses the reviewers concerns and the entire breast cancer cohort enters the follow-up period without recently recorded antibiotic use (and presumable with microbiota intact).

-A related comment relates to the handling of patients with prior antibiotic exposure (which is not infrequent). The latest approach is "Patients using antibiotics in the year prior to diagnosis will be allocated to a past antibiotic use category and remain in this category for the duration of follow-up.". As this past exposure will be a mix of true past exposure (if no further antibiotics were used) and current exposure (with new antibiotic being used), this past exposure group will not be informative, and the study provide no evidence for this group of patients. Again, in my view a better more detailed exposure classification is very much preferred (which would reflect the complexity of the antibiotic exposures and uncertain causal pathways).

*We agree with the reviewer that the past exposure category in the main analysis may not be informative. Consequently, we propose excluding this category from the analysis. This will have no impact on the main analysis which compares antibiotic users after diagnosis with non-users after diagnosis restricted to breast cancer patients who did not use antibiotics in the year before diagnosis.

However, it is worth noting that those using antibiotics in the year before diagnosis will be investigated in the following analyses:

- 1) In the registered report – in the first additional analysis we will repeat the main analysis not excluding individuals using antibiotics in the year before diagnosis and in the second additional analysis we will repeat the main analysis adjusting for use in the previous year.
- 2) Breast cancer patients using antibiotics before diagnosis will be compared with patients not using antibiotics before diagnosis in an additional analysis (moved out of the stage 1 report after the first round of reviewer/editorial comments).
- 3) We will add an additional analysis (not in the registered report) in which we will compare antibiotic user with non-users after diagnosis restricted to individuals on antibiotics in the year before diagnosis.

Reviewer #4 (Remarks to the Author):

I've been asked to comment specifically on the design table and whether the sensitivity analysis should be seen as exploratory analysis and thus removed entirely from the table. The authors addressed all my previous comments sufficiently.

For the design table. The authors should edit the table to remove the mention of exploratory analysis since it only includes main and sensitivity analyses. As far as whether to include the sensitivity analyses, I'd recommend keeping them included since they can be specified in advanced, which they are, including how to interpret the results. Opposed to unplanned analyses, which can not, by definition, but are good to state that there is interest in exploring these relationships during stage 1 of the Registered Report.

*In line with the reviewer's comments we have retained the sensitivity analyses.

Editorial Note: STAGE 2

Dear Editor,

We thank the reviewers for taking the time to review our paper. We have responded to their comments below and made changes to the manuscript which are highlighted using track changes.

We hope the new version of the manuscript is now suitable for publication in Nature Communications but we are happy to make further changes.

Best wishes,

Chris

Prof Chris Cardwell

REVIEWERS' COMMENTS

Reviewer #2 (Remarks to the Author):

The study report will be of significance to the field as there are few studies that have considered the association of antibiotic use in women with breast cancer on breast-cancer specific mortality.

The authors refer to Unplanned analyses (Stage 2 analyses) which involve sensitivity analysis around the definition of outcome, and are reported separately in the results section.

The conclusions appear appropriate - the authors do not over interpret the significance of the findings as they suggest residual confounding might still exist.

There are some minor comments for clarification and that might improve the final manuscript.

Line 31-32: The aim in abstract refers to 'we investigated whether breast cancer patients using oral antibiotics had increased breast cancer-specific mortality', however, the results also report on all-cause mortality (which is not the main analysis but described as sensitivity analysis in the methods).

*We think this provides important context but we would remove it at the discretion of the editor.

Page 8 – line 187 please insert 'pooled' before 'adjusted hazard ratio'

*We have corrected.

Line 218 'patients prescribed 12 or more antibiotics was reduced' – perhaps refer to the revised HR or % risk.

* As recommended by the reviewer to quantify this better we now state:

"This increased risk in patients prescribed 12 or more antibiotics was reduced in sensitivity analyses adjusting for infections to 44%, and also reduced in sensitivity analyses limiting reverse causality (by applying longer lags or restricting to antibiotics in the 5 years after diagnosis)."

Line 230-231 when referring to other studies, the authors state 'no significant association with any antibiotic use after diagnosis (HR=1.39 95% CI 0.93, 2.29),' was shown even though the HR is quite large – this is likely due to the small study size, perhaps the authors refer to the strengths or limitations of these studies by way of critique.

*We have now altered this sentence to incorporate the weakness that this was a small study. We now state: "A recent study of 772 triple negative breast cancer patients did not observe a significant association with any antibiotic use after diagnosis (HR=1.39 95% CI 0.93, 2.29), which could reflect the study size, but observed an increase in breast cancer-specific mortality with increasing numbers of antibiotic prescriptions (HR per prescription=1.05 95% CI 1.01, 1.08)."

The paper mentioned by one reviewer Domzaridou et al was not mentioned in the discussion?

*We agree this study is relevant so we have added the following line:

"Another study, from the UK, observed a 36% (adjusted HR=1.36 95% CI 1.23, 1.49) increased risk of all-cause mortality in antibiotic users compared with non-users, but is more difficult to directly compare with our findings because it investigated antibiotic use in the three month period before breast cancer diagnosis rather than after diagnosis."

Discussion limitations – what about the fact that over the counter medications are not captured? Also, there were no implications of the findings mentioned in the discussion – is it possible to discuss the consequences or application of the findings to practice or policy?

*Our main exposure antibiotics is not available over the counter so we do not think that missing over the counter medications will have a large influence on our findings.

*As highlighted later, the interpretation of our findings is complicated and it is difficult to make a clear recommendation on the basis of our findings.

Reviewer #2 (Remarks on code availability):

The Stata code provide was not in the form of a README file, and limited comments but with the appropriate 'cohort' file available and variables listed it should be possible to run (I did not attempt to run the code on the data). The models provided in the code were (i) a model with only antibiotic use and (ii) a model with antibiotic use (presumably y/n) and a list of covariates. The description of the 'antibyn' variable and other covariates used were not described in detail in the accompanying table.

*We have added further description to the do file. We think the code could easily be cut and paste or typed into a STATA do file and then run.

Reviewer #3 (Remarks to the Author):

I thank the authors for this work.

My comments on Journal's questions:

-Whether the data are able to test the authors' proposed hypotheses by passing the approved outcome-neutral criteria (such as absence of floor and ceiling effects or success of positive controls)

=> yes

- Whether the Introduction, rationale and stated hypotheses are the same as the approved Stage 1 submission (I have appended a copy of the in-principle accepted Stage 1 manuscript and Supplementary Information)

=> yes

- Whether the authors adhered to the registered experimental procedures

=> Important to state when analyses were conducted: were all analyses conducted after registration of the protocol or were some analyses conducted before?

*Yes we can confirm this and we state: "At the time of writing the Stage 1 Registered Report, all of the data or evidence that was used to answer the research question existed but was inaccessible to the research team (consistent with a level 5 Registered Report)."

- Whether any unregistered exploratory analyses added by the authors are justified, methodologically sound, and informative

=> OK

- Whether the authors' conclusions are justified given the data

=>The conclusion states that "the attenuation of the associations in sensitivity analyses, and similar findings for other causes of death, suggest this increase may be attributable to residual confounding". I am not sure about the strength of this conclusion. Firstly, the sensitivity analyses around the effects of other causes of death ignore the real challenges of using cause-of-death data to attribute exact of cause of death. Death certificates tend to overrepresent immediate complications. This analysis of death due to other causes lacked a robustness of e.g. negative control design. Secondly, several sensitivity analyses seem to support the primary hypothesis (e.g. change of lag time), so conclusion is too categorical. Thirdly, it is well known that the more subgroup analyses one conducts (e.g. found in randomised trials), the muddier the results become. Fourthly, Thirdly, there is the fundamental question of what to conclude if primary and relevant sensitivity analyses vary. The conclusion, I think, is that no firm conclusions are possible.

*We agree with the reviewer it is hard to reach conclusions and in support of this we clearly state in the manuscript:

"The cause of the increased breast cancer-specific mortality in patients prescribed 12 or more antibiotics after diagnosis is unknown."

We also state:

"This association is consistent with animal studies that suggest that repeated antibiotic use could have a detrimental carcinogenic impact on the microbiome, reducing beneficial microbiota, and resulting in increased risk of breast cancer recurrence and death."

It is hard to convey this in the abstract due to the word limit but we think that the current abstract is sufficiently balanced, and the other reviewers appear to agree, but we would revise it further at the discretion of the editor.

Additional comment:

=>Good to conduct literature search on previous work that has explored the same research question. For example, I could find no reference and discussion of related work: DOI: 10.3390/currenol30090614

*We thank the author. We agree this study is relevant and we have added the following line:

“Another study, from the UK, observed a 36% (adjusted HR=1.36 95% CI 1.23, 1.49) increased risk of all-cause mortality in antibiotic users compared with non-users, but is more difficult to directly compare with our findings because it investigated antibiotic use in the three month period before breast cancer diagnosis rather than after diagnosis.”

Reviewer #3 (Remarks on code availability):

No expert in stata code

Reviewer #4 (Remarks to the Author):

The authors do a very nice job of describing their results from their stage 1 Registered Report, including any deviations and unplanned analyses. My only comment is in the introduction - the authors might want to end of introduction it might be good to change the tense (e.g., “The study will investigate...” to ‘The study does investigate....’)

*Thanks, we have changed tense as requested.